# Prominent misinformation interventions reduce misperceptions but increase scepticism

Emma Hoes ®¹✉, Brian Aitken ®², Jingwen Zhang ®³, Tomasz Gackowski ®⁴ & Magdalena Wojcieszak ®³

Current interventions to combat misinformation, including fact-checking, media literacy tips and media coverage of misinformation, may have unintended consequences for democracy. We propose that these interventions may increase scepticism towards all information, including accurate information. Across three online survey experiments in three diverse countries (the United States, Poland and Hong Kong; total $n$ = 6,127), we tested the negative spillover effects of existing strategies and compared them with three alternative interventions against misinformation. We examined how exposure to fact-checking, media literacy tips and media coverage of misinformation affects individuals' perception of both factual and false information, as well as their trust in key democratic institutions. Our results show that while all interventions successfully reduce belief in false information, they also negatively impact the credibility of factual information. This highlights the need for further improved strategies that minimize the harms and maximize the benefits of interventions against misinformation.

Scholars, observers and policymakers worry that information that is false, fabricated, untrustworthy or unsubstantiated by credible evidence can have dramatic consequences for democracy. These worries are sparked by recent events feared to be triggered by misleading claims. For instance, Trump tweeting that the 2020 election was rigged allegedly mobilized his supporters and led to what we now know as the Capitol Riots on 6 January 2021. Similarly, false information claiming that COVID vaccines are harmful may have led to clusters of communities refusing effective vaccines for pandemic mitigation[1,2]. Some research suggests that continued exposure to misinformation may lead to lasting misperceptions as a result of increased familiarity with factually inaccurate information[3–5]. Other work goes as far as saying that misinformation can influence political behaviour or election outcomes[6]. Because of such fears, institutions, agencies, platforms and scholars have directed plentiful resources to determining how to fight

misinformation and make citizens more resilient. Combined efforts have resulted in well-known and established intervention strategies—namely, fact-checking, media literacy, and news media covering and correcting misinformation. These interventions are hoped to counter the spread of and belief in misinformation.

From 2016 to 2018 alone, an estimated 50 independent fact-checking organizations were established[7], and numerous news outlets incorporated fact-checking practices as part of their business, such as the 'Reality Check' page on the BBC's website, the *New York Times*'s 'Fact Checks' and CNN's 'Facts First'. Media literacy interventions also burgeoned[8–10]. These aim to prevent rather than correct the potential impact of misinformation by educating the public on how to critically evaluate the quality of information[9]. Examples of this are Facebook's '10 Tips to Spot Fake News' and professional (journalistic) training programmes offered by the growing organization First Draft.

¹Department of Political Science, University of Zurich, Zurich, Switzerland. ²Huron Consulting Group, Chicago, IL, USA. ³Department of Communication, University of California, Davis, Davis, CA, USA. ⁴Department of Communication and Public Relations, University of Warsaw, Warsaw, Poland. ✉e-mail: hoes@ipz.uzh.ch

Finally, news media organizations increased their coverage of misinformation more generally with the aim of raising awareness about its prevalence and effects. News media, whether partisan or not, increasingly focus on fake news and misinformation. Plotting the occurrence of the terms 'fake news', 'misinformation' and 'disinformation' in major US newspapers over time (as archived by LexisNexis) and the frequency of people searching for these terms on Google Search and Google News in the United States[11] show a remarkable increase in the popularity of these terms, starting around the 2016 US presidential election. For example, whereas there were 1,979 articles mentioning one of the terms in 2010, there were 9,012 such articles in 2017.

How effective are these strategies against misinformation? On the one hand, some studies suggest that fact-checking shows promising effects depending on the timing[12], the source[13] and the kinds of labels used[14,15]. On the other hand, fact-checking alone is deemed insufficient to correct misperceptions[16] or, in some cases, is even shown to be counterproductive such that it can reinforce inaccurate beliefs[6,17,18]. In a similar vein, media literacy courses and general warnings about the presence of misinformation can also have unintended spillover effects. Such spillover effects (that is, the unintended consequences for democracy of interventions against misinformation) may make people critical towards not only misinformation but also factually accurate information[14,19,20]. Furthermore, recent research looking beyond misperceptions as an outcome finds that news media's attention to misinformation decreases trust in science and politics[11].

Accordingly, this project addresses an overarching question of theoretical and practical importance: how can we improve interventions against misinformation to minimize their negative spillover effects? We rely on pre-registered online survey experiments (see https://osf.io/t3nqe for the full pre-registration administered on 22 August 2022) in three countries—the United States, Poland and Hong Kong—to offer a comprehensive test of both positive and negative effects of fact-checking, media literacy interventions and the coverage of misinformation, side by side. We argue that the reason that these interventions may generate misperceptions, scepticism towards verified facts, and political and institutional distrust has to do with the way the message is delivered, as detailed below. For instance, strategies against misinformation often adopt a negative tone (for example, "Fight the Fake" or "Proceed with Caution"), put blame on political actors and media outlets, or amplify the harms of misinformation. We thus additionally propose applicable ways to prevent these negative effects from emerging. We compare existing delivery strategies of fact-checking, media literacy efforts and media coverage of misinformation with alternatives that incorporate theoretically driven adjustments to existing interventions. For coherence and parsimony, we focus on the effects of each strategy on three main outcomes: misperceptions, scepticism and trust. We report the results for the remaining pre-registered outcomes in Supplementary Section A.3 and refer the reader to our pre-registration for the rationale of these remaining outcomes.

For fact-checking, one common approach is to put emphasis on the (political) actor making inaccurate claims, which can be the originator of a false claim (for example, a politician) or the medium that spreads it (for example, a news outlet). The main fact-checking organizations in the United States, such as PolitiFact and Snopes, often explicitly and visibly name the 'source' of the fact-checked claim. For instance, PolitiFact's fact-checked claims are accompanied by a logo (in the case of (social) media) or picture (in the case of public figures) and the name of the source: for example, "Robert F. Kennedy, Jr. stated on July 11, 2023 in a video…". This approach—which we call the Accountability Strategy—may negatively affect people's trust in politicians and media (the sources of false claims) because it explicitly blames the spreader of misinformation[21]. In addition, fact-checking efforts emphasizing the accountable actor may accidentally foster misperceptions by increasing the ease of (mis)information retrieval and the familiarity of the claim[22–26].

Many (recent) media literacy efforts—including those coming from big social media companies such as Twitter and Facebook—typically focus on how to spot misinformation[20]. A good and prominent example of this is Facebook's '10 Tips to Spot Fake News'. We call this the Misinformation Focus. Although this strategy may be successful at helping people identify inaccurate claims[27,28] by triggering accuracy motivations[29], it may also generate negative spillover effects and increase scepticism towards otherwise true or factual pieces of information[20]. In addition, this strategy can decrease trust in various democratic actors (for example, (social) media, scientists and journalists) by emphasizing that it is difficult to know whom and what to trust.

Furthermore, news media coverage of misinformation in general and of false claims in particular often repeats misinformation and emphasizes its existence, spread and threats[30], without putting these threats in the necessary context. That is, by repeating false claims, news media inadvertently increase the reach of these claims, and by giving disproportionate attention to misinformation, news media may generate the perception that misinformation is prevalent. This runs counter to recent empirical evidence suggesting that exposure to and effects of misinformation and untrustworthy sources are very limited[31–36]. We call such coverage the Decontextualized Approach and suspect that it may have unintended consequences such that it decreases trust and increases scepticism towards verified facts[11,20]. Moreover, by repeating falsehoods, news coverage of misinformation may come with a risk of fostering misperceptions by making them more familiar[26], more easily retrievable from memory and therefore easier to process[25].

In addition to identifying the effects of these existing strategies, we examine how they can be improved to prevent undesired spillover effects (that is, decreased trust, increased scepticism and inaccurate beliefs) and maximize positive outcomes (that is, preventing misperceptions). We propose that fact-checkers should consider what is important to emphasize when addressing (mis)information: the source (Accountability Strategy) or the verification of the relevant claim (which we call the Correctability Strategy). This approach relies on fact-checkers' and journalists' expertise in issue and frame selection, in which they engage as part of their daily practice[37,38], and calls for an assessment of the need for Accountability versus the need for Correctability. More specifically, we suggest that—when appropriate—putting more emphasis on the claim itself (as opposed to emphasizing the source) might overcome negative spillover effects on trust and misperceptions. By focusing on the correctability of a claim, fact-checkers can emphasize evidence-based thinking without attributing blame to a politician or a news source. As pre-registered, we hypothesize that the Correctability Strategy will lead to lower levels of both misperceptions (H1$_a$) and scepticism (H1$_b$), as well as higher levels of trust (H1$_c$) than the Accountability Strategy.

To improve existing media literacy interventions, we propose that they should not limit their attention to misinformation (the Misinformation Focus)[16] but also focus on detecting partisan bias, which we call the Bias Focus. After all, news bias and hyper-partisan reporting are more prevalent and also bigger problems for various democratic processes than misinformation[39]. This should help citizens better evaluate the quality of information in general while at the same time reducing the negative effects on scepticism and trust by not overemphasizing the role of misinformation in the news media ecosystem. Both strategies have in common that they should trigger accuracy motivations[29], making people invest more cognitive resources in problem-solving and analytical thinking[27,28] and thus helping people recognize misinformation. Yet, only the Bias Focus should help identify misinformation without increasing scepticism towards all information, for several reasons.

Because this strategy specifically teaches people to identify balanced legacy media on top of biased media, it should help individuals identify accurate information. In addition, the Bias Focus should minimize scepticism towards all information because it may prompt people to think about how information is presented and framed.

**Table 1 | Definitions of existing and proposed strategies**

| Intervention | Existing strategy | Definition | Proposed strategy | Definition |
|---|---|---|---|---|
| Fact-checking | Accountability Strategy | Focus on source of claim (the actor) | Correctability Strategy | Focus on content of claim |
| Media literacy | Misinformation Focus | Tips to spot 'fake news' | Bias Focus | Tips to spot news biases |
| Coverage of misinformation | Decontextualized Approach | No reference to scope conditions of misinformation | Contextualized Approach | Reference to scope conditions |

These tips should encourage people to evaluate the underlying assumptions and motivations behind news stories. This nuanced thinking should enhance media consumers' ability to identify not only reliable information or overt misinformation but also subtler forms of manipulation, such as selective reporting or framing. Furthermore, focusing on biases highlights the importance of context in news reporting. Media literacy interventions can teach individuals to look for multiple sources and perspectives to gain a more comprehensive understanding of an issue. This can lead to the discovery of different viewpoints, without necessarily making individuals sceptical of all information. Lastly, the Bias Focus still encourages critical thinking, but it shifts the emphasis from outright distrust to informed scepticism. Participants learn to evaluate news stories on the basis of factors such as evidence, source credibility and logical coherence. Taken together, we propose that the Bias Focus empowers individuals to make more informed judgements when consuming media. In line with these arguments, we hypothesize that participants exposed to the Bias or Misinformation Focus will be less likely to endorse misperceptions (H2$_a$), but those exposed to the Bias Focus will have lower levels of scepticism (H2$_b$) and higher levels of trust (H2$_c$) than participants exposed to the Misinformation Focus. Note that with these hypotheses, we deviate from the pre-registration in two ways. First, we had additionally formulated hypotheses on the identification of false, accurate and biased news using different measurements. For the sake of parsimony and coherence, in the main paper we only focus on three main outcomes (trust, scepticism and misperceptions) of interest for each of the independent variables, but we present the results of these hypothesized outcomes in Supplementary Information. Second, in the pre-registration we formulated no directional hypothesis on the effect of the Bias Focus and the Misinformation Focus on misperceptions, but we include it here to be able to show the effect of each treatment on the same outcomes, and in the direction based on the rationale outlined above. Note that we thus formulated this hypothesis after pre-registration but before data analysis.

Finally, to counteract the negative spillover effects of media coverage of misinformation, we propose that when covering misinformation, journalists should give context to the issue. In addition to informing about misinformation or a particular falsehood (that is, raising awareness), media should put it in the context of the most recent and best available scientific research. Currently, such evidence points out that—given the limited exposure to[34] and effects of[35] misinformation—misinformation is not as grave a problem as is often suggested. We thus compare the Contextualized Approach (that is, covering the problem in its broader context) to the Decontextualized Approach (that is, news media coverage without the context). While in both approaches news media may still report a particular falsehood to correct it and raise public awareness, putting the false claim in a wider context may deflate as opposed to inflate the salience of such a claim. We predict that participants exposed to the Contextualized Approach will have higher levels of trust (H3$_a$) and lower levels of misperceptions (H3$_b$) and scepticism (H3$_c$) than participants in the Decontextualized Approach.

An overview of all the discussed strategies and their definitions can be found in Table 1. To systematically isolate the causal effects of our treatments, we opted for subtle differences between existing strategies to fight misinformation versus our proposed strategies by changing one key component of each strategy. Larger differences between experimental stimuli increase the chances of observing significant differences between treatments, but at the cost of uncertainty of what is driving these differences. Moreover, previous studies have shown that even small differences in experimental stimuli, such as altering a single word[14], can lead to substantial differential effects[40].

To offer a comprehensive portrayal of the effects—positive and negative—of the three existing strategies versus our proposed strategies, we conducted online experiments in the United States (n = 2,008), Poland (n = 2,147) and Hong Kong (n = 1,972). While we did not pre-register any country-specific hypotheses, we selected Poland, Hong Kong and the United States for this survey experiment because these countries represent diverse cultural and political contexts, thus allowing us to explore the generalizability of interventions against misinformation across different societies. We randomized participants to one of the six treatment groups (each strategy makes up one treatment group) or a control group. A visualization of what participants in each treatment condition were presented with can be found in Supplementary Section A. Note that the materials in Supplementary Information have been redacted for legal reasons, but all original materials can be found on OSF at https://osf.io/5xc7k/. As can be seen from the original materials, the treatments simulate realistic social media posts on a professionally designed and interactive social media site.

After exposure to treatment, all participants were asked to rate the accuracy of several true (measuring scepticism) and false (measuring misperceptions) claims as well as indicate their levels of trust in various institutions (for example, journalists, social media, traditional media and scientists). To test our hypotheses, we compared each proposed strategy to its corresponding existing strategy and the control group. Our key take-away is that most interventions against misinformation—including our proposed strategies to improve fact-checking, media literacy and the coverage of misinformation—come at a cost. While some seem to effectively decrease misperceptions, these interventions at the same time increase scepticism in true information. This means that people tend to rate not only false but also accurate information about important political topics as unreliable, untrustworthy and inaccurate. These effects are most pronounced in the United States and Poland, but less so in Hong Kong, where the effects are largely insignificant, although the coefficients follow similar directions. Given the pronounced dominance of true or trustworthy content over false or untrustworthy content in (social) media environments, this is a worrisome trend.

## Results

We began by assessing our key question of interest: compared with existing interventions to fight misinformation, do our proposed strategies overcome negative spillover effects on scepticism in verified facts, misperceptions and trust? We used the statistical software program R (version 2022.12.0+353; https://posit.co/download/rstudio-desktop/) to estimate pre-registered (see https://osf.io/t3nqe for the full pre-registration), two-tailed ordinary least squares regressions with the assigned treatment as the independent variable, the three outcomes as dependent variables and all the pre-treatment variables as covariates (that is, demographics; see Supplementary Section B.2 for the full list of covariates included in the analysis). While not pre-registered, we additionally clustered standard errors at the respondent level on the basis of reviewer request. The clustering did not change any of the

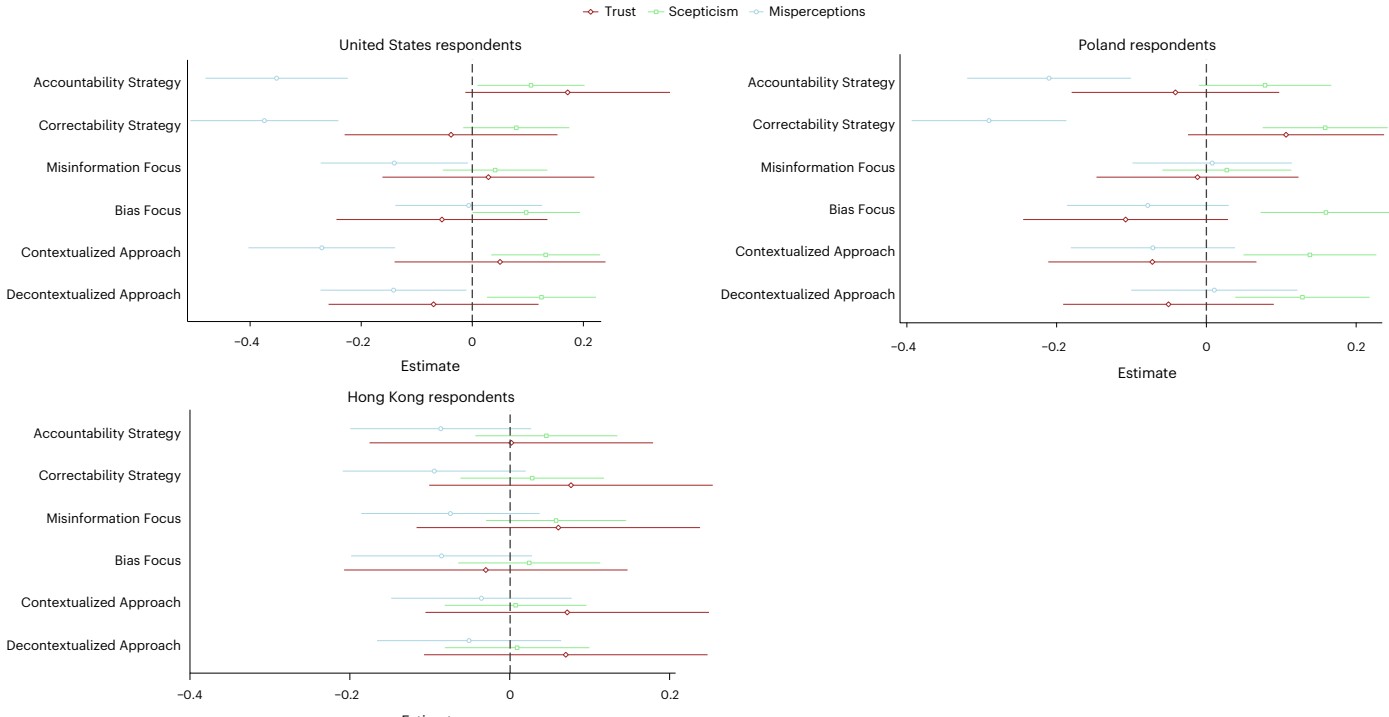

**Fig. 1 | Misperceptions, scepticism and trust outcomes by treatment and country.** The misperception (light blue), scepticism (light green) and trust (brown) coefficient estimates for each of the six treatment conditions by country are shown. The error bars represent 95% CIs. The fully reported results can be found in Supplementary Section A.1. For the United States, see Supplementary Tables 4–6. For Poland, see Supplementary Tables 22–24. For Hong Kong, see Supplementary Tables 40–42. For the United States, there were 2,008 participants over seven independent experiments; for Poland, 2,147 participants over seven independent experiments; and for Hong Kong, 1,972 participants over seven independent experiments.

| | Accountability Strategy | Correctability Strategy | Misinformation Focus | Bias Focus | Contextualized Approach | Decontextualized Approach |
|---|---|---|---|---|---|---|
| Reduces misperceptions | ✓ ✓ ✗ | ✓ ✓ ✗ | ✓ ✗ ✗ | ✗ ✗ ✗ | ✓ ✗ ✗ | ✓ ✗ ✗ |
| Increases scepticism | ✓ ✗ ✗ | ✓ ✓ ✗ | ✗ ✗ ✗ | ✓ ✓ ✗ | ✓ ✓ ✗ | ✓ ✓ ✗ |
| Affects trust | ✗ ✗ ✗ | ✗ ✗ ✗ | ✗ ✗ ✗ | ✗ ✗ ✗ | ✗ ✗ ✗ | ✗ ✗ ✗ |

United States | Poland | Hong Kong

**Fig. 2 | Findings by country.** The checks represent support for hypotheses, whereas the Xs represent unsupported hypotheses.

results. Also on the basis of reviewer request, we computed a discernment measure that subtracts the perceived accuracy of false claims from that of true claims. These results can be found in Supplementary Section C.4. Following our pre-analysis plan, we did these analyses separately for the United States, Poland and Hong Kong. The Methods section provides the details on the data and methodology, including the exact measurement of our dependent variables.

## General effects
Figure 1 shows the effects of the three interventions (fact-checking, media literacy and media coverage of misinformation), comparing the existing and our proposed strategies to the control group for each of the three countries. Figure 2 provides an overview of all effects. Supplementary Section C.1 contains detailed regression output of all the results. Almost all interventions—including our proposed strategies—are successful at reducing misperceptions (measured as the perceived accuracy of false claims), with some minor (mostly statistically insignificant, but see A5 for the Bayes factor (BF) analyses) differences between the different strategies (see Supplementary Tables 4 and 22 for the fully reported results for the United States and Poland, respectively). However, the same interventions also increase

scepticism (measured as the perceived accuracy of true claims; Supplementary Tables 5 and 23). This is largely true for the United States and Poland, but not Hong Kong, where none of the interventions yield statistically significant effects (Supplementary Table 40; see Supplementary Table 91 for BFs), although the coefficients move in similar directions. In none of the countries did most of the interventions affect people's general trust (see Supplementary Table 6 for the United States and Supplementary Table 24 for Poland) or trust in any specific institutions (using the individual items) (see Supplementary Tables 7–13 for the US results and Supplementary Tables 25–31 for Poland). Furthermore, in almost all cases, we found no statistically significant differences between the existing and our proposed strategies (see Supplementary Tables 14–21 for the US results, Supplementary Tables 32–39 for Poland and Supplementary Table 91 for BFs). These results hold when running the analyses only on those who passed the manipulation check (Supplementary Section C.2). It seems, however, that the mentioned effects, whether positive (reduced misperceptions) or negative (increased scepticism), are fairly short-lived. In the United States, we administered a follow-up survey one week after exposure to the treatments. We found that all the effects disappeared (Supplementary Tables 64 and 65). We also rejected all trust-related

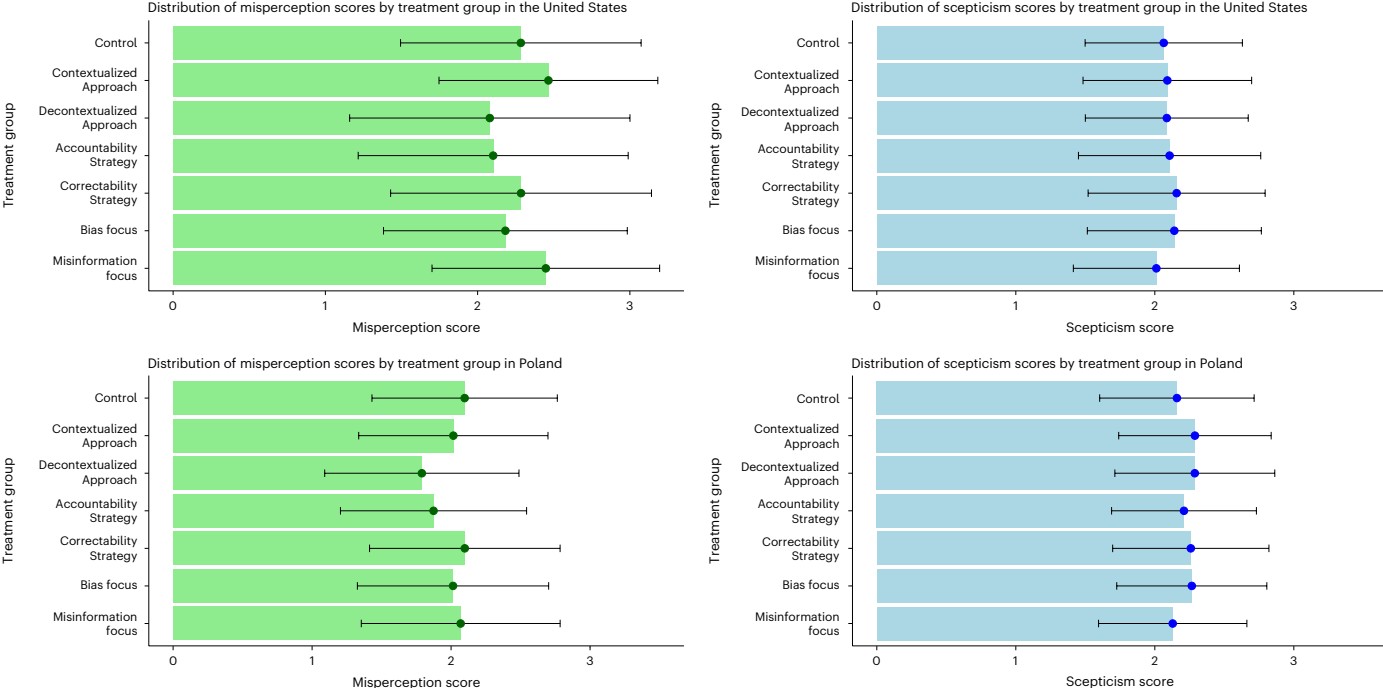

**Fig. 3 | Mean misperception and scepticism by treatment group in the United States and Poland.** The mean misperception scores (in green) and the mean scepticism scores (in blue) across all seven independent experiment treatments (United States: $n = 2,008$; Poland: $n = 2,147$) are shown. Misperception scores were constructed by averaging the respondents' accuracy ratings of false statements, whereas scepticism scores were created by averaging the respondents' accuracy ratings of true statements. The data are presented as mean values ± s.d.

hypotheses and relegate the presentation of the results for Hong Kong to Supplementary Section C.1.3.

## Effects per country

We next turned to a more detailed discussion of our results per country (see Fig. 3 for a visualization of the mean levels of scepticism and misperception per treatment group and country). In the United States, all strategies except the Bias Focus decreased misperceptions compared with the control group (Accountability Strategy: $\beta = -0.352$; $P = 0.001$; 95% confidence interval (CI), −0.48, −0.22; Correctability Strategy: $\beta = -0.374$; $P = 0.001$; 95% CI, −0.51, −0.24; Misinformation Focus: $\beta = -0.14$; $P = 0.038$; 95% CI, −0.27, −0.01; Contextualized Approach: $\beta = -0.2709$; $P = 0.001$; 95% CI, −0.40, −0.14; Decontextualized Approach: $\beta = -0.142$; $P = 0.034$; 95% CI, −0.27, −0.01).

However, in the United States, the Contextualized Approach ($\beta = 0.132$; $P = 0.008$; 95% CI, 0.04, 0.23), the Decontextualized Approach ($\beta = 0.125$; $P = 0.0121$; 95% CI, 0.03, 0.22), the Accountability Strategy ($\beta = 0.106$; $P = 0.029$; 95% CI, 0.01, 0.2), and the Bias Focus ($\beta = 0.1$; $P = 0.05$; 95% CI, 0, 0.19) all increased scepticism relative to the control group. This means that most tested strategies came with the negative spillover effect of making citizens more sceptical towards true and verified information.

We additionally found that only the two fact-checking strategies improved discernment between the false and true claims (that is, the subtractive measure; Supplementary Table 88), whereas none of the media literacy or media coverage strategies did. We further reflect on this finding in Discussion.

We see a similar trend in Poland. Compared with the control group, all strategies except for the Misinformation Focus and the Accountability Strategy increased scepticism (Contextualized Approach: $\beta = 0.138$; $P = 0.002$; 95% CI, 0.05, 0.23; Decontextualized Approach: $\beta = 0.128$; $P = 0.005$; 95% CI, 0.04, 0.22; Correctability Strategy: $\beta = 0.159$; $P = 0.001$; 95% CI, 0.08, 0.24; Bias Focus: $\beta = 0.16$; $P = 0.001$; 95% CI, 0.07, 0.25). There are again no statistically significant differences between the corresponding strategies. In Poland, both fact-checking strategies,

the Correctability Strategy ($\beta = -0.29$; $P = 0.001$; 95% CI, −0.39, −0.17) and the Accountability Strategy ($\beta = -0.21$; $P = 0.001$; 95% CI, −0.32, −0.1), were successful at reducing misperceptions. In sum, just like in the United States, in Poland most tested interventions also came with the negative spillover effects of making citizens more sceptical towards true and verified information, while—in the case of Poland—having only minimal positive effects of reducing misperceptions.

We also found that the Decontextualized Approach decreased the ability to discern between false and true claims (that is, the subtractive measure; Supplementary Table 89). All other strategies did not significantly affect discernment in Poland. We further reflect on these findings in Discussion.

Finally, two of our covariates—political interest and age—predicted our outcomes strongly and fairly consistently across all treatment groups and in both the United States (Supplementary Section C.1.1) and Poland (Supplementary Section C.1.2). Political interest is negatively correlated with scepticism towards true information in both the United States ($\beta = -0.086$; $P = 0.001$; 95% CI, −0.12, −0.06) and Poland ($\beta = -0.044$; $P = 0.001$; 95% CI, −0.07, −0.02). It also predicts decreases in misperceptions in Poland ($\beta = -0.036$; $P = 0.03$; 95% CI, −0.07, −0.003), while predicting increases in misperceptions in the United States ($\beta = 0.054$; $P = 0.001$; 95% CI, 0.02, 0.09). Older participants in both countries (United States: $\beta = -0.119$; $P = 0.001$; 95% CI, −0.14, −0.09; Poland: $\beta = -0.059$; $P = 0.001$; 95% CI, −0.08, −0.04) hold lower misperceptions than younger participants, with these effects being greater in the US sample. In the United States, age ($\beta = 0.1$; $P = 0.001$; 95% CI, 0.08, 0.12) also increases scepticism towards true information; a similar effect for Poland misses the traditional threshold for statistical significance ($\beta = 0.014$; $P = 0.08$; 95% CI, −0.001, 0.03).

## Discussion

This project delves into the potential negative consequences of current strategies to fight misinformation: fact-checking, media literacy tips and media coverage of misinformation. Online survey experiments in three diverse countries—the United States, Poland and

Hong Kong—examined how these three strategies impacted individuals' perception of both inaccurate and factual information as well as trust in key democratic institutions.

While dominant interventions, such as fact-checking, media literacy tips and news coverage of misinformation, aim to prevent the spread and endorsement of misinformation, we found that they may inadvertently prime individuals to approach all information, whether false or true, with heightened suspicion and scepticism. This is concerning, as this suggests that mere exposure to alarming labels such as 'misinformation' or 'fake news' in the media and public discourse may lead to negative consequences and reduce people's trust in verified information. These findings, along with similar recent evidence[20,41], suggest that existing misinformation mitigation approaches need to be redesigned.

To address these challenges, we proposed three alternative strategies for fact-checking, media literacy tips and media coverage of misinformation, and compared their effects to those of the existing approaches. We thus aimed to offer systematic evidence on a key question of relevance to the current societal climate: how can existing interventions be improved such that they do not reduce trust, increase scepticism or foster misperceptions? Answering this question and identifying which messaging and delivery strategies of fact-checking, media literacy and coverage of misinformation can minimize the harms while maximizing the benefits of these interventions could offer practical guidelines for media organizations, the educational sector and policymakers.

Our results demonstrate that while the tested interventions are successful at reducing belief in false information, they simultaneously increase scepticism in the credibility of factual information. This is the case not only for existing strategies but also for our proposed 'improved' strategies. In other words, individuals who are exposed to all these interventions may become more likely to perceive true information as untrustworthy or inaccurate. This is particularly alarming given the prevalence of the discussion of and interventions against misinformation in today's media ecosystem. Public discourse about misinformation has the potential to prime individuals to be excessively distrustful of all information, a phenomenon that has been observed in previous research[11,20].

Given that the average citizen is very unlikely to encounter misinformation[32,33], wide and far-reaching fact-checking efforts or frequent news media attention to misinformation may incur more harms than benefits. Put differently, because most people are much more likely to encounter reliable news than misinformation, any increase in general scepticism may have a much stronger negative effect than the positive effect of reducing misperceptions[42]. While one could still argue that an increase in scepticism may be worth the positive effect of decreased misperceptions as long as it improves discernment between true and false information, our additional analyses indicate that the majority of strategies, apart from fact-checking within the US context, have an indeterminate impact on discernment. This suggests that there is insufficient evidence to conclusively determine their effects on enhancing the discernment between truth and falsehood. Decontextualized coverage of misinformation even worsened discernment in Poland. This latter finding in particular underscores that the benefits of reduced belief in false information may not outweigh the negative consequences of decreased belief in accurate and reliable news. In essence, the potential gains from reducing misperceptions must be carefully weighed against the broader implications of heightened scepticism in our information landscape.

There are several potential explanations for the limited evidence of differences in effects between the existing and our proposed strategies. First, the differences between our treatments were subtle: more than one third of our sample failed the manipulation check. This indicates that these differences were not noticeable to many participants. We note, however, that our interventions did not yield different effects

among participants who did pass the manipulation check. Nevertheless, it is important to acknowledge that our BF analyses indicate moderate evidence against the null hypothesis of no differences between the existing and proposed strategies, suggesting that our results should be interpreted with caution (see Supplementary Table 91 for BF values). We encourage future work to both strengthen and run more experiments with our adapted interventions. Scholars could design strategies that differ more substantially from existing strategies. This way it may be harder to determine what aspect of the intervention is driving the effects, but more substantial changes to existing interventions against misinformation are arguably needed to maximize their benefits and limit their harms.

Second, our findings raise the question of whether the public effectively distinguishes between concepts such as falsehoods and bias, especially given that the Bias Focus in media literacy did not affect scepticism or misperceptions. As detailed in Supplementary Table 91, additional BF analyses present a nuanced picture: in Poland, the evidence leans towards our interventions having a notable effect, while in the United States, the evidence is less conclusive or even suggests minimal impact. This variation indicates that cultural or contextual factors might play an important role in how media literacy interventions are perceived and their effectiveness. Because a part of the public may equate partisan or biased news with 'fake news', the differential impact of our interventions could have been obscured. This underlines the importance for scholars, journalists and educators to develop strategies that not only elucidate the differences between falsity and bias but also provide practical tools for identifying and evaluating these elements in diverse real-world contexts.

Third, our Contextualized news coverage treatment only informed people that misinformation is not widespread. Yet, the extent of misinformation could have been more clearly contextualized. Although many individuals struggle with numerical concepts, treatments that present a numerical anchor may be more impactful. Without such an anchor, people may still overestimate the prevalence and impact of misinformation, even when informed about its limited extent. Still, it is important to acknowledge that there is a delicate balance between pinpointing the appropriate level of manipulation strength in experimental setups and designing treatments that can effectively shape real interventions.

Finally, we offer two potential explanations for the (lack of) findings for Hong Kong (see Supplementary Section C.5 for BFs). First, the Hong Kong findings might be due to the sample age difference. We had difficulties in recruiting older adults from the Hong Kong survey vendor. In the Hong Kong sample, only 9.2% of participants were aged 55 and older. This age category was 33.1% in the Poland sample and 31.9% in the US sample. The younger sample in Hong Kong might have more baseline media literacy and experiences with misinformation corrections, which could attenuate the intervention effects. Another possible explanation has to do with Hong Kong's political environment. Some scholars argue that the lack of impact of misinformation or misinformation intervention is due to largely steady political inclinations of the Hong Kong population. The overall trend in public opinion regarding whether Hong Kong should be governed under the 'one country, two systems' principle has not changed much since 2014 or even earlier. Thus, regardless of frequent exposures to misinformation and increasing availability of misinformation interventions, the influence of these interventions might be limited[43].

We acknowledge several limitations to our study. First, although survey experiments are a powerful tool for determining causality, they have limited external validity. We aimed to increase the external validity of our design by delivering treatments as realistic social media posts featured on a professionally designed and interactive social media site. Nevertheless, it is possible that our findings do not generalize to real-world situations where individuals are not exposed to interventions against misinformation in such a controlled manner. Second, our

measurement of misperceptions was based on the perceived accuracy of self-fabricated claims. This may limit the ecological validity of our findings. At the same time, relying on existing false claims may have introduced the possibility that participants had encountered these claims prior to the study, which could impact their perceived accuracy. Finally, we offer potential explanations for the (lack of) findings for Hong Kong.

Regardless, these findings highlight the need for caution when attempting to combat misinformation and signal the difficulty of designing interventions that do so. Naturally, it is important to address false information. Yet, when doing so, it is critical to carefully craft and test strategies to not inadvertently erode citizens' trust in accurate information. As scholars, policymakers, educators and journalists navigate the ever-changing (mis)information landscape, it is imperative that we continue to delve into the intricate dynamics between misinformation, public scepticism and the effectiveness of interventions aimed at promoting accuracy and the consumption of verified information. Future research should focus on examining the specific strategies and techniques that best preserve trust in reliable sources while combating falsehoods. Such comprehensive exploration will be instrumental in shaping more targeted and effective approaches to addressing the challenges posed by misinformation in our information-driven society.

## Methods

We conducted online survey experiments in the United States, Poland and Hong Kong. The study in each country was conducted from 30 August 2022 to 27 October 2022. The follow-up survey in the United States was conducted from 8 September 2022 to 5 November 2022. This study received institutional review board approval from the University of California, Davis, approval no. 1792005-2. Participants were recruited using Dynata in the United States, Panel Ariadna in Poland and Qualtrics in Hong Kong. People were compensated by Dynata, Panel Ariadna and Qualtrics directly, and the price per respondent we paid was US$3.25, US$2.50 and US$4.80, respectively. These opinion-polling companies used stratification based on census information on age, gender and education level. Our US dataset included a total of 2,008 participants (mean age, 45 years; 50.22% female; 70.21% white). Our Poland dataset included a total of 2,147 participants (mean age, 45.64 years; 35.61% female). Finally, our Hong Kong dataset included a total of 1,972 participants (mean age, 37.93 years; 43.81% female). We based the sample size on a power analysis using the software G*Power version 3.1 (ref. 44) (see section 3.5 in the pre-registration at https://osf.io/t3nqe for the full calculation and rationale). After giving informed consent to participate in the study (see Supplementary Section B.3 for the consent form), the respondents first completed a pre-survey answering questions about their sociodemographic characteristics, political attitudes and beliefs (see Supplementary Section B.1 for all covariates and their measurement). Next, the participants were presented with the following instructions: "On the next pages you will see two Facebook messages posted by (media) organizations and Facebook users in the last few days. Please read each message carefully. At the end of the survey we will ask you some questions about them. Please note that the posts are not interactive in this part of the study."

Each participant was randomly assigned to one of six treatment conditions or a control group. Each treatment featured two Facebook posts on the top of the newsfeed on a mock Facebook site, with a standardized number of comments and likes across conditions and with the background functions blurred (see Supplementary Section A for a redacted version of all stimulus materials per country, and https://osf.io/5xc7k/ for the original materials). All survey and stimulus materials were translated by native Polish and Cantonese speakers with minor contextual adaptations for the two countries for the claims and specific sources.

Specifically, we kept the media literacy tips consistent across the countries. For the other treatments and study materials, we kept all the statements (in both the treatment texts and the used false/true claims) as close to one another as possible. For instance, the treatment text in the Accountability Strategy started from the same 'skeleton'—for example, the statement "Recently, [ACTOR] claimed that [STATEMENT]" was exactly the same for Poland and Hong Kong. We replaced [ACTOR] with comparable left- or right-leaning politicians in each country. For Poland, we chose known/unknown politicians from the governing coalition and from the opposition. For the Hong Kong stimuli, we similarly chose known/unknown politicians from the Pro-Establishment (also called Pro-Beijing) camp or the Pro-Democracy camp. The contentions between these political factions in Poland and in Hong Kong are similar to the left/right division in the United States. The same parallel approach goes for all true/false claims in the project. For instance, we used the same false claims across all countries (merely changing, for example, "Local government officials in Michigan" to "Local government officials from [a specific region in Poland/Hong Kong]"). This careful selection and adaptation for both the treatment texts and other study materials assures that the politicians are equally known and partisan across the countries, that the sentences have the same baseline level of plausibility, and so forth. The original true and false claims in each language can be found in Supplementary Section A.2.1, and the used 'skeletons' are provided in Supplementary Section A.1.

To increase external validity, the two Facebook posts that made up each treatment were interactive, such that participants could use the range of Facebook reactions to each post (for example, like, love or laugh) as well as comment below them. Naturally, the website did not have the functionality of resharing.

After exposure to treatment, the participants were redirected to the questionnaire measuring the core outcomes (Supplementary Section B.2). Scepticism was measured by asking the participants how accurate they thought three true statements were to the best of their knowledge on a four-point scale (from 1, "Not at all accurate", to 4, "Very accurate"). Misperceptions were measured using the same scale, but about two false statements. The false statements were self-fabricated (that is, made up), and both the true and the false statements were selected from a pre-tested pool of claims that were rated as similarly easy to read, interesting, easy to understand, likely to be true (false), (un)believable, and equally plausible among Democrats and Republicans. Trust was measured by asking the participants to report how much they trusted seven institutions—journalists, scientists, fact-checkers, traditional media, university professors, social media and the government—on a seven-point scale (from 1, "I don't trust it/them at all", to 7, "I completely trust it/them"). For each outcome, we aggregated the items to create one single measure of scepticism, misperceptions or trust. In addition, after exposure to treatment, we presented the participants with a statement serving as a manipulation check (see Supplementary Section B.5 for the item wording). Across the samples, 62.8% of US participants, 69.9% of Polish participants and 62.1% of Hong Kong participants passed the manipulation check. There were no statistically significant differences by demographics (for example, age or education) between those who passed and failed (Supplementary Section C.2). Finally, the respondents were informed about the nature of this study through a debriefing (Supplementary Section B.3.2).

The mean level of scepticism was 2.13 with s.d. 0.68 in the United States, 3.24 with s.d. 1.4 in Poland and 2.28 with s.d. 0.52 in Hong Kong. For misperceptions, this was 2.23 with s.d. 0.86 in the United States, 1.26 with s.d. 1.11 in Poland and 2.32 with s.d. 0.67 in Hong Kong. The mean trust level was 3.81 with s.d. 1.37 in the United States, 2.31 with s.d. 1.89 in Poland and 4.3 with s.d. 1.05 in Hong Kong. We completed a pre-registration for all analyses (https://osf.io/t3nqe). We subsequently realized that our pre-registration plan did not include a hypothesis for the effect of the media literacy treatments on misperceptions and thus included a relevant hypothesis after data collection. All other predictions and analyses were according to our pre-registration.

## Reporting summary

Further information on research design is available in the Nature Portfolio Reporting Summary linked to this article.

## Data availability

The replication data, including all (stimulus) materials used in this study, are available at https://osf.io/5xc7k/.

## Code availability

The replication code is available at https://osf.io/5xc7k/.

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

## Acknowledgements

We thank Y. Cheng and C. Y. Song from Hong Kong Baptist University for their help with translation and data collection in Hong Kong. We acknowledge the support of the following funding sources: Facebook/Meta (Foundational Integrity & Impact Research: Misinformation and Polarization; principal investigator, M.W.; co-principal investigator, E.H.) and the European Research Council, 'Europeans exposed to dissimilar views in the media: investigating backfire effects' Proposal EXPO-756301 (ERC Starting Grant; principal investigator, M.W.). Any opinions, findings and conclusions or recommendations expressed in this material are those of the authors and do not necessarily reflect the views of Facebook/Meta or the European Research Council. The funders had no role in study design, data collection and analysis, decision to publish or preparation of the manuscript.

## Author contributions

Conceptualization: E.H. and M.W. Analyses: E.H. and B.A. Investigation: E.H., B.A., J.Z., M.W. and T.G. Visualization: B.A. and E.H. Writing— original draft: E.H. and M.W. Writing—review and editing: E.H., B.A., J.Z. and M.W.

## Funding

## Competing interests

M.W. and E.H. received funding from Facebook/Meta (Foundational Integrity & Impact Research: Misinformation and Polarization).

Facebook/Meta had no role in study design, data collection and analysis, decision to publish or preparation of the manuscript. B.A. is currently employed by Huron Consulting Group, a management consulting firm. At the time when the research was carried out, B.A. was a PhD student at the University of California, Davis. The other authors declare no competing interests.

## Additional information

**Correspondence and requests for materials** should be addressed to Emma Hoes.

# Reporting Summary

## Statistics

For all statistical analyses, confirm that the following items are present in the figure legend, table legend, main text, or Methods section.

| n/a | Confirmed | |
|---|---|---|
| ☐ | ☒ | The exact sample size (*n*) for each experimental group/condition, given as a discrete number and unit of measurement |
| ☐ | ☒ | A statement on whether measurements were taken from distinct samples or whether the same sample was measured repeatedly |
| ☐ | ☒ | The statistical test(s) used AND whether they are one- or two-sided *Only common tests should be described solely by name; describe more complex techniques in the Methods section.* |
| ☐ | ☒ | A description of all covariates tested |
| ☐ | ☒ | A description of any assumptions or corrections, such as tests of normality and adjustment for multiple comparisons |
| ☐ | ☒ | A full description of the statistical parameters including central tendency (e.g. means) or other basic estimates (e.g. regression coefficient) AND variation (e.g. standard deviation) or associated estimates of uncertainty (e.g. confidence intervals) |
| ☒ | ☐ | For null hypothesis testing, the test statistic (e.g. *F*, *t*, *r*) with confidence intervals, effect sizes, degrees of freedom and *P* value noted *Give P values as exact values whenever suitable.* |
| ☐ | ☐ | For Bayesian analysis, information on the choice of priors and Markov chain Monte Carlo settings |
| ☐ | ☒ | For hierarchical and complex designs, identification of the appropriate level for tests and full reporting of outcomes |
| ☐ | ☒ | Estimates of effect sizes (e.g. Cohen's *d*, Pearson's *r*), indicating how they were calculated |

*Our web collection on statistics for biologists contains articles on many of the points above.*

## Software and code

Policy information about availability of computer code

| | |
|---|---|
| Data collection | Participants were recruited using Dynata in the US, Panel Ariadna in Poland, and Qualtrics in Hong Kong. These opinion polling companies used stratification based on census information on Age, Gender and Education Level. We calculated sample size for power analysis using the G*Power software (version 3.1.9.6). |
| Data analysis | Data was analyzed using the statistical software R (version 2022.12.0+353). The code used to analyze the data can be found at https://osf.io/t3nqe |

For manuscripts utilizing custom algorithms or software that are central to the research but not yet described in published literature, software must be made available to editors and reviewers. We strongly encourage code deposition in a community repository (e.g. GitHub). See the Nature Portfolio guidelines for submitting code & software for further information.

## Data

Policy information about availability of data

All manuscripts must include a data availability statement. This statement should provide the following information, where applicable:
- Accession codes, unique identifiers, or web links for publicly available datasets
- A description of any restrictions on data availability
- For clinical datasets or third party data, please ensure that the statement adheres to our policy

The replication data, including all (stimulus) materials used in this study, is available at https://osf.io/t3nqe

# Human research participants

Policy information about studies involving human research participants and Sex and Gender in Research.

| | |
|---|---|
| Reporting on sex and gender | We introduced a measure of self-reported gender (male/female/other) as a covariate in all our analyses. It was measured asking: "How do you describe yourself? |
| Population characteristics | Our US dataset included a total N = 2008 participants; mean age = 45 years, 50.22% female, 70.21% white. Our Poland dataset included a total of N = 2147 participants; mean age = 45.65 years, 35.61% female. Finally, our Hong Kong dataset included a total of N = 1972 participants; mean age = 37.93 years, 43.81% female. In the Hong Kong sample, we only had 9.2% of participates aged 55 and older. This age category was 33.1% in the Poland sample and 31.9% in the US sample. These differences may introduce biases. |
| Recruitment | Participants were recruited using Dynata in the US, Panel Ariadna in Poland, and Qualtrics in Hong Kong. These opinion polling companies used stratification based on census information on Age, Gender and Education Level. Relying on these polling companies may introduce self-selection bias because individuals who choose to participate in online surveys administered by such companies may differ systematically from the broader population, potentially leading to an unrepresentative sample that does not accurately reflect the attitudes and characteristics of the target populations in the US, Poland, and Hong Kong. |
| Ethics oversight | This study received IRB approval from the University of California Davis, approval no: 1792005-2 |

Note that full information on the approval of the study protocol must also be provided in the manuscript.

# Field-specific reporting

Please select the one below that is the best fit for your research. If you are not sure, read the appropriate sections before making your selection.

☐ Life sciences   ☒ Behavioural & social sciences   ☐ Ecological, evolutionary & environmental sciences

For a reference copy of the document with all sections, see nature.com/documents/nr-reporting-summary-flat.pdf

# Behavioural & social sciences study design

All studies must disclose on these points even when the disclosure is negative.

| | |
|---|---|
| Study description | Quantitative Online Survey Experiment |
| Research sample | Participants were recruited using Dynata in the US, Panel Ariadna in Poland, and Qualtrics in Hong Kong. These opinion polling companies used stratification based on census information on Age, Gender and Education Level, and is representative on these demographic variables. Our US dataset included a total N = 2008 participants; mean age = 45 years, 50.22% female, 70.21% white. Our Poland dataset included a total of N = 2147 participants; mean age = 45.65 years, 35.61% female. Finally, our Hong Kong dataset included a total of N = 1972 participants; mean age = 37.93 years, 43.81% female. We selected Poland, Hong Kong, and the US for this survey experiment as these countries represent diverse cultural and political contexts, thus allowing us to explore the generalizability of interventions against misinformation across different societies. |
| Sampling strategy | Stratified sample (see above). We used the software program G*Power to conduct a power analysis. Our goal was to obtain .80 power to detect a small effect size of 0.02 at the standard 0.05 alpha error probability. We also use a Bonferroni adjustment by dividing the nominal alpha level, 0.05, by the maximum number of comparisons we could make (e.g., not only between the variations of each type of intervention and the control group, but also between types of interventions, and using all the outcomes for each intervention). |
| Data collection | All data was collected online by the professional opinion polling companies. Data collection and analyses were not performed blind to condition |
| Timing | The study in each country was conducted from 8/30/22 to 10/27/22. The follow-up survey in the US was conducted from 9/8/22 to 11/5/22. |
| Data exclusions | Participants who failed the attention check were replaced by the opinion polling companies in each country. This means we eventually did not have to exclude any data from our analyses. |
| Non-participation | No participants dropped our of declined participation |
| Randomization | Each participant was randomly assigned to one of six treatment conditions of a control group |

# Reporting for specific materials, systems and methods

We require information from authors about some types of materials, experimental systems and methods used in many studies. Here, indicate whether each material, system or method listed is relevant to your study. If you are not sure if a list item applies to your research, read the appropriate section before selecting a response.

| Materials & experimental systems | | Methods | |
|---|---|---|---|
| **n/a** | **Involved in the study** | **n/a** | **Involved in the study** |
| ☒ ☐ | Antibodies | ☒ ☐ | ChIP-seq |
| ☒ ☐ | Eukaryotic cell lines | ☒ ☐ | Flow cytometry |
| ☒ ☐ | Palaeontology and archaeology | ☒ ☐ | MRI-based neuroimaging |
| ☒ ☐ | Animals and other organisms | | |
| ☒ ☐ | Clinical data | | |
| ☒ ☐ | Dual use research of concern | | |

