## [Peer Review File · Nature Human Behaviour]

Peer Review Information

Journal: Nature Human Behaviour

Manuscript Title: Prominent Misinformation Interventions Reduce Misperceptions but Increase Skepticism

Corresponding author name(s): Emma Hoes

Reviewer Comments & Decisions:

Decision Letter, initial version:

9th August 2023

Dear Emma,

Thank you once again for your manuscript, entitled "Prominent Misinformation Interventions Reduce Misperceptions but Increase Skepticism," and for your patience during the peer review process. I am sorry for the delay in reaching a decision.

Your manuscript has now been evaluated by 4 reviewers, whose comments are included at the end of this letter. Although the reviewers find your work to be of interest, they also raise some important concerns. We are interested in the possibility of publishing your study in Nature Human Behaviour, but would like to consider your response to these concerns in the form of a revised manuscript before we make a decision on publication.

To guide the scope of the revisions, the editors discuss the referee reports in detail within the team, including with the chief editor, with a view to (1) identifying key priorities that should be addressed in revision and (2) overruling referee requests that are deemed beyond the scope of the current study. We hope that you will find the prioritised set of referee points to be useful when revising your study. Please do not hesitate to get in touch if you would like to discuss these issues further.

In particular, we ask that you address the following (as well as all other reviewer comments):

- 1) Address reviewer concerns regarding the presentation and rationale for your interventions, ensuring that they are conceptually clear and well-motivated.
- 2) Address all reviewer technical concerns, including about the success of the manipulation check, clustering of standard errors, and the choice of modelling approach.
- 3) Carry out the additional analyses of discernment suggested by Reviewers 3 and 4.

In sum, we invite you to revise your manuscript taking into account all reviewer and editor comments. We are committed to providing a fair and constructive peer-review process. Do not hesitate to contact us if there are specific requests from the reviewers that you believe are technically impossible or unlikely to yield a meaningful outcome.

We hope to receive your revised manuscript within two months. I would be grateful if you could contact us as soon as possible if you foresee difficulties with meeting this target resubmission date.

- Include a "Response to the editors and reviewers" document detailing, point-by-point, how you addressed each editor and referee comment. If no action was taken to address a point, you must provide a compelling argument. When formatting this document, please respond to each reviewer comment individually, including the full text of the reviewer comment verbatim followed by your response to the individual point. This response will be used by the editors to evaluate your revision and sent back to the reviewers along with the revised manuscript.
- Highlight all changes made to your manuscript or provide us with a version that tracks changes.

[REDACTED]

We look forward to seeing the revised manuscript and thank you for the opportunity to review your work. Please do not hesitate to contact me if you have any questions or would like to discuss these revisions further.

Sincerely,
[REDACTED]

Reviewer expertise:

Reviewer #1: sociology, misinformation

Reviewer #2: communication, misinformation

Reviewer #3: political psychology, fact-checking

Reviewer #4: political behaviour, misinformation

REVIEWER COMMENTS:

Reviewer #1:

Remarks to the Author:

I think this is an excellent work, exploring experimentally a point overlooked in research on misinformation. I have only minor comments that could improve the presentation.

Is Figure 1 necessary? I would suppose readers of NHB are quite familiar with the explosion of misinformation research/coverage. What is described in the text could be sufficient.

The point about fact-checking seems one-sided. While I agree with the authors that there is an interesting difference between blaming the source and focusing on the claim, I am not sure that, in general, existent fact-checking is mostly about the former, as implied in the first paragraph of the section "Existing Strategies to Fight Misinformation". I would suggest modifying this paragraph.

The presentation of the results could benefit by remaining closer to the hypotheses presented before (H1a, H1b, etc.). Perhaps a table summarising the results in this way could help readers.

I wonder whether it would be possible to do an exploratory analysis to assess which strategy works better overall. E.g., in line with the model from Acerbi, A., Altay, S., & Mercier, H. (2022), cited in the manuscript, one could parametrise (roughly) the amount of misinformation VS reliable news "in the real world" and see how the interventions proposed (in particular the effects on misconceptions and scepticism) affect the overall quality of information believed. The idea is that there is an asymmetry in the effects: the increase in general scepticism has a much stronger negative effect than the positive effect of reducing misperceptions, as people are much more likely to encounter reliable news than misinformation. From this perspective, how would be the six interventions rated? If a proper exploratory analysis goes beyond the scope of the paper, it could be discussed anyway.

Penultimate line of page 11: "the f complex dynamics": typo.

Reviewer #2:

Remarks to the Author:

Thank you for giving me the opportunity to review this manuscript. It is concerned with the question of how current and newly developed misinformation interventions affect misinformation beliefs and information skepticism as well as trust, utilizing a pre-registered experimental online study in three countries, Hong Kong, Poland, and the U.S.

The study is timely, addresses an important research question, and specifically focuses on finding solutions that can help contain the impact of misinformation while trying to minimize the risk for the general information environment, based on findings from previous research that can show that misinformation interventions can increase general trust.

I enjoyed reading how traditional misinformation intervention strategies were further developed by the authors. There is a certain logic to 'correctability', bias focus', and 'contextualization', and I consider it worth trying out these strategies when countering misinformation. However, I would very much like to know more about the theoretical background that the effectiveness of each strategy builds on. I understand that there is a space restriction here. However, I would still ask the authors to shortly explain why the focus on biases instead of misinformation would reduce misinformation beliefs while at the same time not increasing general media skepticism. Right now, the selection is somewhat eclectic, and including stronger theoretical accounts could help readers to understand better the results in terms of why the effect of interventions did not play out as expected.

Related to this, I find the role of the 'actor' in explaining the 'accountability strategy' ambiguous. Are we talking about the originator of a false claim or the media outlet that spreads it? Both can be a source and depending on the case, it can either be the originator (e.g., a politician) or the medium (e.g., Breitbart) that becomes the center of the coverage and is mentioned as a 'spreader' of misinformation.

The study is methodologically rigorous and follows the pre-registration plan by and large. I still have a few questions:

- a) The pre-reg plan mentions a manipulation check, which is also part of the supplementary material, but the results of this check are not supported. Was the manipulation check successful?
- b) Why, despite using random assignment of participants to treatment conditions, were OLS regressions used? The analytical strategy does not state why OLS regression was deemed superior compared to, for example, ANOVA analysis.
- c) I would like to know to what extent the mock Facebook site used was interactive, as stated on p. 11.
- d) Looking at the manipulations in detail, I wondered why a "Check your biases" post was still part of the misinformation focus media literacy intervention and not the 'bias focus' intervention. In other words, how distinct are the two manipulations?

The results are presented clearly and stringently. They help us to assess the effectiveness of misinformation strategies.

In the discussion, reasons for the findings should be more in focus. The authors mention the possibility that the manipulations were 'too weak', which is why my question about the manipulation check is important to address. Should this not have been successful, I think we have a somewhat clear indication that these light, realistic changes in the stimuli do not work as they should, which does not disregard the effectiveness of the proposed strategies. It's a thin line between finding the right manipulation strength in an experimental setting and the right framing of actual interventions. This challenge could be focused more strongly in the discussion section. My take is that delivering a one-fits-all message to audiences seems to be difficult when it comes to this topic, and more personalized approaches, taking news experiences, cognitive skills, and topic involvement could lead to more

hopeful results.

Good luck with your work!

Reviewer #3:

Remarks to the Author:

This paper makes a number of key contributions to the literature on misinformation correction. First, it cleanly delineates between multiple types of misinformation interventions (fact-checking vs. media literacy interventions vs. increasing the salience of misinformation). Second, it denotes flaws with existing correction approaches that may lead people exposed to them to just display reduced trust in all claims rather than just false ones, and in their place the paper proposes new tactics. However, in a large-scale, multi-country survey experiment, the paper finds that virtually all approaches simultaneously reduce the perceived accuracy of both true and false information.

While the main conclusion of the paper is not fully novel (the authors already cite other examples of papers that find misinformation correction reduces belief in both true and false information), the paper is useful in that tests between disparate strands of approaches to misinformation correction. Furthermore, even though it attempts to remedy issues with prior misinformation correction methods, the same pattern holds up - fighting misinformation tends to make people more skeptical of EVERYTHING. As a result, this paper can serve as an effective "call to arms" for subsequent research to figure out how to help consumers of information separate the informational wheat from the chaff. These findings are of broad public interest because belief in false and misleading information runs rampant, and even if people are not widely exposed to false news, these claims can travel independently of media consumption (Druckman, Levendusky, & McLain, 2018). Despite my positive impression of the manuscript, though, there are a number of points this manuscript should address in order to maximize its true potential.

THEORY:

- The authors present the accountability strategy of outfits such as PolitiFact and Snopes as focused on the source of the claim, presenting their content-focused fact-checks in contrast to such an approach. However, such a presentation of what these outfits do might be exaggerated. Ultimately, fact-checking websites address claims even when they discuss the source of such claims. A more productive way of discussing the two types of fact-checks instead seems to be on what to emphasize, rather than which information is present.

RESEARCH DESIGN AND RESULTS:

- The fact that the authors use such a diverse sample of countries is a major strength of the paper. However, it also presents some challenges for interpreting results. One major challenge comes from differences in treatment information across countries. The media literacy tips are consistent across samples. However, the claims that are being fact-checked (and also mentioned in the "coverage of misinformation" treatments) differ across samples. Are these claims comparable enough? For example, are they equally contentious across countries (which implicates political motivated reasoning)? Are they the subject of similar levels of baseline plausibility?

- With very little exception, in a subsequent wave of the US experiment, none of the treatments have any effects on misperceptions, skepticism, and trust (Table A12). One could argue that this means

both the intended (misperception reduction) and unintended consequences (increases skepticism) are fairly short-lived.

- The experiments are clear that almost all methods of misperception correction reduce the perceived accuracy of both true and false claims. However, one common method of examining the effects of intervention is using truth discernment - the perceived accuracy of true relative to false headlines (Pennycook & Rand, 2021). Even if the treatments have the unintended result of making people more skeptical even when they shouldn't be, do they still improve truth discernment?
- The theoretical reasons for switching from a focus on falsity to a focus on bias make sense. However, given that doing so does not change the overall effect of exposure to media literacy material, it does raise the question over whether the public meaningfully distinguishes falsity from bias. This is worth discussing in the conclusion section.
- Similarly, one limitation of how the authors switched from a de-contextualized to a contextualized treatment highlighting the prevalence of false news is that there is no addition of a numerical anchor of fake news exposure. Since most people struggle with numeracy and intuitively processing small, but non-zero percentages, it may be that even when people are told that fake news isn't a huge problem, that they are still greatly overestimating its prevalence and impact. While I don't think this is a fatal flaw in the paper, it might be a reason why contextualizing seems not to do much.

PRESENTATION:

- The authors share several figures in the Appendix of the treatments. However, they are quite small and difficult to read. Since supplemental information can basically be of unlimited length online, there seems to be no need to keep them so small. Enlarging them would make it easier for attentive readers to follow.
- The stimulus materials included appear to only be the ones in the US experiment. However, it would be worth adding the materials for the Polish and Hong Kong experiments.
- While there are explanations for the preregistered outcomes in the OSF pre-registration for Tables A12-A24 in the Appendix, they are not present in the actual manuscript. Even if the authors do not wish to mention much of these results in analysis, some explanation within the manuscript would be helpful. That said, I do think Table A12 could be worth a mention as evidence that both intended and unintended consequences of misinformation correction are short-lived. Tables A13/A15 (US), A17/A19 (Poland), and A21/A23 (Hong Kong) provide useful evidence that the media literacy treatments, in practice, don't seem to help people with actually identifying false and biased headlines (respectively) in real time.

REFERENCES:

Druckman, J.N., Levendusky, M.S., & McLain, A. (2018). No Need to Watch: How the Effects of Partisan Media Can Spread via Interpersonal Discussions. *American Journal of Political Science*, 62(1), 99-112. <https://dx.doi.org/10.1111/ajps.12325>

Pennycook, G., & Rand, D.G. (2021). The psychology of fake news. *Trends in Cognitive Sciences*, 25(5), P388-P402. <https://doi.org/10.1016/j.tics.2021.02.007>

Reviewer #4:

Remarks to the Author:

I was glad to have the opportunity to review this paper. The authors identify potential problems with

existing approaches to tackling the problem of online misinformation, propose alternatives, and test both empirically.

- I agree with the authors' assessment that judging misinformation interventions solely on their ability to decrease belief in (or sharing of) fake content is problematic, as they may also decrease belief in (or sharing of) real content. Given that there is far more real content than fake content in the information ecosystem, this appears extremely problematic. In general, I believe this is the strongest theoretical prediction and empirical result. Much of the paper is concerned with testing the authors' concerns about specific misinformation intervention approaches and their proposed alternatives. I'm more skeptical of these critiques and proposed solutions.

1. Accountability vs. Correctability

- I understand how deemphasizing the source of the misinformation could lead to less distrust of news sources, but I don't think the authors are making a strong enough case in the current draft. There are currently only a few sentences on this in the paper, and I can't read the stimuli examples in the appendix due to the low resolution of the figures.

2. Misinformation vs. Bias Focus

- On "Bias focus": "This should help citizens better evaluate the quality of information in general while at the same time reducing the negative effects on skepticism and trust by not overemphasizing the role of misinformation in the news media ecosystem." But wouldn't this Bias Focus similarly decrease trust and increase skepticism by emphasizing bias? This seems problematic.

- How does Bias Focus increase accuracy motivations? This doesn't make sense to me. What is the mechanism there?

- Relatedly, this line ---"Yet, only the 'Bias Focus' strategy should help identify accurate information") ---confuses me...Are you saying that Misinformation Focus interventions do not help people to identify accuracy information. This is perplexing.

3. De-Contextualized vs. Contextualized

- On a similar note, the contextualized alternative approach is confusing to me. My understanding of the research that the authors recommend journalists reference when covering misinformation shows that misinformation is not as large of a problem as commonly thought because it comes from a relatively small number of posters and is seen by relatively few people on social media. Yet the influence of misinformation can be far broader than this. It is covered by the media, spread through other channels, and can enter into political discourse (note that this paper begins by detailing the many ways in which misinformation is dangerous).

- Moreover, covering a piece of misinformation (e.g., microchips in vaccines) and then telling readers that concerns about misinformation are overblown seems like an odd way to cover misinformation, particularly given uncertainty about how overblown concerns about misinformation really are. Isn't this akin to telling people to get Covid booster shots while also mentioning that they aren't always deemed necessary for large segments of the population?

- One further point on the contextualized approach. You're suggesting that news outlets report on a piece of misinformation and then tell readers that concerns about misinformation are overblown, in part because not many people are exposed to misinformation. But by doing this, you're exposing people to misinformation. Given the illusory truth research that you reference, isn't this problematic?

- Guay & colleagues have a short paper, forthcoming in Nature Human Behavior, on a similar topic. They advocate for measuring belief in both true (i.e., real) and false (i.e., fake) content, partially for the reasons you lay out in this paper. They also advocate for using a measurement of discernment as the primary outcome of interesting. Essentially, the treatment effect on discernment captures how much more people in the treatment group believe true vs. false content than those in the control group. Right now you report the constitute parts of discernment (belief in false, disbelief in true), but really what you care about is whether the difference between them is significant, right? That might be worth considering.
- This paper contains a lot of labels, and they are sometimes hard to keep track of (the table helps with this). The accountability and correctability labels were hard to keep track of as a first time reader. The other labels are more self-explanatory.
- I can't see Figure 2 clearly enough to tell difference between the interventions. I'm having the same issue for the stimuli figures in the appendix
- Do you cluster standard errors, either at the respondent level (each respondent sees multiple headlines) and/or the headline level (each headline is seen by multiple respondents). The latter is only necessary if the same headlines were not seen by all respondents (that is, if all respondents saw the same X headlines, this isn't an issue). But either way you need to account for the nested structure of the data.
- There are some small spelling and grammatical errors throughout. Another careful read-through may be helpful.
- It would be nice to briefly mention why some strategies may be harmful in the introduction (even in a single sentence), rather than teasing them. The current version reads "We argue that the reason that these interventions may generate misperceptions, skepticism toward verified facts, and political and institutional distrust has to do with the way the message is delivered, as detailed below."
- I'm confused by this sentence: "After all, some evidence suggests that exposure to and effects of misinformation and untrustworthy sources are limited". Is your point just that exposure to misinfo doesn't really have much of an effect, so reporting on it in the news might have a worse effect? If so, that's not coming through. Also, if so, I think you need to elaborate on the mechanism a bit. Is the idea that misinfo has a limited effect in the wild because people aren't very exposed to it, and that when news outlets cover it people are exposed to it more than they would otherwise have been?
- You spend a lot of real estate in the paper explaining country-specific findings. Why? Can you motivate this a bit more?

Very small comments:

- "democratic consequences" in abstract and intro reads funny. Consequences for democracy?
- "participants exposed to the bias and Misinformation Focus will be less likely to endorse
- Misperceptions". I had to read this part of the paper a few times, before realizing that there is no condition that exposes people to both the Misinformation Focus and Bias Focus interventions. Also, it gets confusing when you use labels for some things (e.g., Misinformation Focus, in this sentence) but not others ("but those receiving a media literacy treatment focused on news bias"). Consistent labelling and capitalization helps the first time reader who is trying to remember which labels refer to what.
- Figure 3 is the main plot, but is quite small. What if you vertically arrange the facets? Just an idea.
- Missing figure captions in most figures

Author Rebuttal to Initial comments**Reviewer #1:**

Remarks to the Author:

I think this is an excellent work, exploring experimentally a point overlooked in research on misinformation. I have only minor comments that could improve the presentation.

We thank the reviewer for this positive feedback!

Is Figure 1 necessary? I would suppose readers of NHB are quite familiar with the explosion of misinformation research/coverage. What is described in the text could be sufficient.

We agree with the reviewer. Therefore, we have kept the text describing the trends and omitted the figure.

The point about fact-checking seems one-sided. While I agree with the authors that there is an interesting difference between blaming the source and focusing on the claim, I am not sure that, in general, existent fact-checking is mostly about the former, as implied in the first paragraph of the section "Existing Strategies to Fight Misinformation". I would suggest modifying this paragraph.

We thank the reviewer for raising this point which was also mentioned by Reviewer 3. In the section on 'Existing strategies to fight misinformation', we have toned down the language to better reflect that putting emphasis on the source is simply one approach to fact-checking, rather than the dominant one. Similarly, we reworded the subsequent section on 'Improved strategies to fight misinformation'. Instead of saying that fact-checkers could shift focus from the accountable actor to the debunking of the claim, we now state that fact checkers should put *more emphasis* on the debunking vis-a-vis the accountable. This, in our view, better reflects that this is a choice of emphasis/framing that fact-checkers can easily and feasibly make.

The presentation of the results could benefit by remaining closer to the hypotheses presented before (H1a, H1b, etc.). Perhaps a table summarising the results in this way could help readers.

We thank the reviewer for this suggestion and have included a table summarising the results under

the 'General Effects' section. Given that our hypotheses were formulated such that we expected a difference between the existing and our proposed strategies, and we found none, virtually all hypotheses can only be partially accepted (they affect our outcomes, but no differences between strategies). For that reason, the table we now include summarizes whether each intervention (per country) reduces misperceptions, increased skepticism, or affects trust. While this table thus more or less contains the same information as Figure 2, we believe the table now gives readers a straightforward reading of our results right away. Additionally, we have mentioned more clearly in the results section that we overall find no difference between strategies.

I wonder whether it would be possible to do an exploratory analysis to assess which strategy works better overall. E.g., in line with the model from Acerbi, A., Altay, S., & Mercier, H. (2022), cited in the manuscript, one could parametrise (roughly) the amount of misinformation VS reliable news "in the real world" and see how the interventions proposed (in particular the effects on misconceptions and scepticism) affect the overall quality of information believed. The idea is that there is an asymmetry in the effects: the increase in general scepticism has a much stronger negative effect than the positive effect of reducing misperceptions, as people are much more likely to encounter reliable news than misinformation. From this perspective, how would be the six interventions rated? If a proper exploratory analysis goes beyond the scope of the paper, it could be discussed anyway.

We agree with the reviewer that this is a crucial point which we had not developed enough in the paper. In agreement with the editor(s), we believe a proper exploratory analysis indeed goes beyond the scope of the paper and thank the reviewer for acknowledging that. However, in the discussion section of the manuscript we have now further emphasized this asymmetry of effects. The relevant part of that section now reads: *"Given that the average citizen is very unlikely to encounter misinformation, wide and far-reaching fact-checking efforts or frequent news media attention to misinformation may incur more harms than benefits. Put differently, because most people are much more likely to encounter reliable news than misinformation, any increase in general skepticism may have a much stronger negative effect than the positive effect of reducing misperceptions"*.

Penultimate line of page 11: "the f complex dynamics": typo.

We thank the reviewer(s) for their close read of our manuscript and have fixed all typos and other (grammatical/spelling) errors.

Reviewer #2:

Remarks to the Author:

Thank you for giving me the opportunity to review this manuscript. It is concerned with the question of how current and newly developed misinformation interventions affect misinformation beliefs and information skepticism as well as trust, utilizing a pre-registered experimental online study in three countries, Hong Kong, Poland, and the U.S.

The study is timely, addresses an important research question, and specifically focuses on finding solutions that can help contain the impact of misinformation while trying to minimize the risk for the general information environment, based on findings from previous research that can show that misinformation interventions can increase general trust.

I enjoyed reading how traditional misinformation intervention strategies were further developed by the authors. There is a certain logic to 'correctability', bias focus', and 'contextualization', and I consider it worth trying out these strategies when countering misinformation.

We thank the reviewer for this positive assessment of our manuscript!

However, I would very much like to know more about the theoretical background that the effectiveness of each strategy builds on. I understand that there is a space restriction here. However, I would still ask the authors to shortly explain why the focus on biases instead of misinformation would reduce misinformation beliefs while at the same time not increasing general media skepticism. Right now, the selection is somewhat eclectic, and including stronger theoretical accounts could help readers to understand better the results in terms of why the effect of interventions did not play out as expected.

We thank the reviewer for their suggestion to expand on our theoretical motivations for the (improved) interventions. We have done so — to the extent possible (word limit) — for each strategy. You can find these edits highlighted in blue in the document. We also agree that especially the Misinformation/Bias Focus needed more explanation, as this seems to be the least straightforward one out of the three (as also noted by the other reviewers). To this end, we have included an additional paragraph outlining several (theoretical) arguments why the bias focus should minimize the increase in skepticism. This section now reads:

“Yet, only the 'Bias Focus' strategy should help identify accurate information without increasing skepticism towards all information for several reasons.

Because this strategy specifically teaches people to identify balanced or objective legacy media on top

of biased media, it should help individuals identify accurate information. In addition, the Bias Focus should minimize skepticism towards all information because it may prompt individuals to think about how information is presented and framed. These tips should encourage people to evaluate the underlying assumptions and motivations behind news stories. This nuanced thinking should enhance media consumers' ability to identify not only reliable information or overt misinformation but also subtler forms of manipulation, such as selective reporting or framing. Furthermore, focusing on biases highlights the importance of context in news reporting. Media literacy interventions can teach individuals to look for multiple sources and perspectives to gain a more comprehensive understanding of an issue. This can naturally lead to the discovery of different viewpoints, without necessarily making individuals excessively skeptical of all information. Lastly, the Bias Focus still encourages critical thinking, but it shifts the emphasis from outright distrust to informed skepticism. Participants learn to evaluate news stories based on factors like source credibility, evidence, and logical coherence. Taken together, we propose that the Bias Focus empowers individuals to make more informed judgments while consuming media.} In line with the arguments set out above, we hypothesize that participants exposed to the Bias \textit{or} Misinformation Focus will both be less likely to endorse misperceptions ($H2\$_{a}\$$), but only those exposed to the Bias Focus will have lower levels of skepticism ($H2\$_{b}\$$) and higher levels of trust ($H2\$_{c}\$$) than participants exposed to the Misinformation Focus.

Related to this, I find the role of the ‘actor’ in explaining the ‘accountability strategy’ ambiguous. Are we talking about the originator of a false claim or the media outlet that spreads it? Both can be a source and depending on the case, it can either be the originator (e.g., a politician) or the medium (e.g., Breitbart) that becomes the center of the coverage and is mentioned as a ‘spreader’ of misinformation.

We thank the reviewer for raising this point and agree that we did not detail what we mean by the accountable ‘actor’. In the section on existing strategies to fight misinformation we now explicitly address this point. Our argument here is that an emphasis on the source (which can be both a politician as well as a medium) can decrease trust in these actors (e.g., Trump, Breitbart) by blaming them. We believe journalists/fact-checkers should make a judgment as to whether and in which scenarios it is more important to keep the actor accountable or correct the claim. This is a choice of emphasis/framing that fact-checkers can easily and feasibly make. We have also highlighted this point more clearly now in the manuscript (see also response to Reviewer 1’s first comment). We hope that the revised paragraph now meets the reviewer’s suggestion.

The study is methodologically rigorous and follows the pre-registration plan by and large. I still have a few questions:

a) The pre-reg plan mentions a manipulation check, which is also part of the supplementary material, but the results of this check are not supported. Was the manipulation check successful?

We thank the reviewer for this important comment that we had overlooked. We now more information about the success of our manipulation. That is, we now provide the percentage of participants who passed the manipulation check in each of the countries (US: 62.8%, PL: 69.9%, and HK: 62.1%). We provide demographic comparison tables of compliers and non-compliers in the Appendix (no significant differences). We also discuss the implications of these results in the discussion-section (highlighted in blue).

b) Why, despite using random assignment of participants to treatment conditions, were OLS regressions used? The analytical strategy does not state why OLS regression was deemed superior compared to, for example, ANOVA analysis.

Thank you for the good question! Indeed, when randomization to experimental conditions is successful (as it was in our project), ANOVA analyses may be sufficient. We clarify that we used OLS regressions because our outcomes are continuous variables and also because OLS regression allows us to include multiple covariates in our analyses. Adjusting for covariates can increase precision in the linear model and thus the precision of the estimated treatment effect. We hope that this explains the rationale well.

c) I would like to know to what extent the mock Facebook site used was interactive, as stated on p. 11.

We apologize for omitting this information and have now included it under the Materials and Methods section: “the two Facebook posts that made up each treatment were interactive such that participants could use the range of Facebook reactions to each post (e.g., like, love, laugh, etc.) as well as comment below them. Naturally, the website did not have the functionality of resharing.”

d) Looking at the manipulations in detail, I wondered why a “Check your biases’ post was still part of the misinformation focus media literacy intervention and not the ‘bias focus’ intervention. In other words, how distinct are the two manipulations?

For the Misinformation Focus media literacy interventions our aim was to stay as close to the original infographic as possible. The graphics and the information were originally published by the

International Federation of Library Associations and Institutions (IFLA), in 2017, and have been spread on social media widely and translated into multiple languages. We slightly amended some of the texts on the original infographic so that the Bias Focus and Misinformation Focus would at the very least have a similar layout/tone, while isolating/manipulating specific differences which could potentially be driving any effects. We apologize for not including this information before, and have now added it to Section A2 under Stimulus Materials in the Appendix. The ‘check your biases’ or ‘reflect on your biases’ for that reason is present in both treatments, albeit somewhat differently phrased. In the method section as well as in the discussion we now note that the differences in our treatments may have been too subtle to generate pronounced effects, and we discuss the implications of that in more detail. In addition, we now include more information on the manipulation-check to directly address the perceived distinction of the two manipulation. We hope that the reviewer finds these explanations and corresponding adaptations suitable and sufficient, and thank the reviewer for raising this question.

The results are presented clearly and stringently. They help us to assess the effectiveness of misinformation strategies.

Thank you! Following a suggestion from Reviewer 1, we have now additionally included an instructive table summarizing the results/hypotheses.

In the discussion, reasons for the findings should be more in focus. The authors mention the possibility that the manipulations were ‘too weak’, which is why my question about the manipulation check is important to address. Should this not have been successful, I think we have a somewhat clear indication that these light, realistic changes in the stimuli do not work as they should, which does not disregard the effectiveness of the proposed strategies. It’s a thin line between finding the right manipulation strength in an experimental setting and the right framing of actual interventions. This challenge could be focused more strongly in the discussion section. My take is that delivering a one-fits-all message to audiences seems to be difficult when it comes to this topic, and more personalized approaches, taking news experiences, cognitive skills, and topic involvement could lead to more hopeful results.

We thank the reviewer for pointing out this important issue and - again - we apologize for having omitted some crucial information on the manipulation check before. This is now included in the manuscript. That is, we now provide the percentage of participants who passed the manipulation check in each of the countries (US: 62.8%, PL: 69.9%, and HK: 62.1%). We provide demographic comparison tables of compliers and non-compliers in the Appendix (no significant differences).

In the discussion, we additionally integrate a more nuanced mention of the implications of the results of the manipulation check, the strength of our manipulations, and the thin balance between ecological and internal validity.

Good luck with your work!

Reviewer #3:

Remarks to the Author:

This paper makes a number of key contributions to the literature on misinformation correction. First, it cleanly delineates between multiple types of misinformation interventions (fact-checking vs. media literacy interventions vs. increasing the salience of misinformation). Second, it denotes flaws with existing correction approaches that may lead people exposed to them to just display reduced trust in all claims rather than just false ones, and in their place the paper proposes new tactics. However, in a large-scale, multi-country survey experiment, the paper finds that virtually all approaches simultaneously reduce the perceived accuracy of both true and false information.

While the main conclusion of the paper is not fully novel (the authors already cite other examples of papers that find misinformation correction reduces belief in both true and false information), the paper is useful in that tests between disparate strands of approaches to misinformation correction.

Furthermore, even though it attempts to remedy issues with prior misinformation correction methods, the same pattern holds up - fighting misinformation tends to make people more skeptical of EVERYTHING. As a result, this paper can serve as an effective "call to arms" for subsequent research to figure out how to help consumers of information separate the informational wheat from the chaff. These findings are of broad public interest because belief in false and misleading information runs rampant, and even if people are not widely exposed to false news, these claims can travel independently of media consumption (Druckman, Levendusky, & McLain, 2018). Despite my positive impression of the manuscript, though, there are a number of points this manuscript should address in order to maximize its true potential.

We thank the reviewer for this encompassing positive feedback summarizing our contributions.

THEORY:

- The authors present the accountability strategy of outfits such as PolitiFact and Snopes as focused on the source of the claim, presenting their content-focused fact-checks in contrast to such an approach.

However, such a presentation of what these outfits do might be exaggerated. Ultimately, fact-checking websites address claims even when they discuss the source of such claims. A more productive way of discussing the two types of fact-checks instead seems to be on what to emphasize, rather than which information is present.

We thank the reviewer for raising this point which was also mentioned by Reviewer 1. In the section on ‘Existing strategies to fight misinformation’, we have toned down the language to better reflect that putting emphasis on the source is simply one approach to fact-checking, rather than the dominant one. Similarly, we reworded the subsequent section on ‘Improved strategies to fight misinformation’. Instead of saying that fact-checkers could shift focus from the accountable actor to the debunking of the claim, we now state that fact checkers should put *more emphasis* on the debunking vis-a-vis the accountable. This, in our view, better reflects that this is a choice of emphasis/framing that fact-checkers can easily and feasibly make.

RESEARCH DESIGN AND RESULTS:

- The fact that the authors use such a diverse sample of countries is a major strength of the paper. However, it also presents some challenges for interpreting results. One major challenge comes from differences in treatment information across countries. The media literacy tips are consistent across samples. However, the claims that are being fact-checked (and also mentioned in the "coverage of misinformation" treatments) differ across samples. Are these claims comparable enough? For example, are they equally contentious across countries (which implicates political motivated reasoning)? Are they the subject of similar levels of baseline plausibility?

We thank the reviewer for raising this important point. Please know that in addition to keeping the media literacy tips consistent across the countries, we kept all the other statements (both in treatments and the false statements) as close to one another as possible. For instance, all the statements in the accountability strategy started from the same ‘skeleton’ (e.g., the statement “Recently, Republican/Democrat [ACTOR] claimed that [STATEMENT]. ...” was exactly the same for Poland, with the exception for “[ACTOR], a politician from the left/right ...” and instead of Known Republican/Democrat and Unknown Republican/ Democrat, we selected Known Governing majority/Opposition politicians and those from Unknown Governing majority/Opposition. The exact same sentence skeleton was used for the Hong Kong stimuli, where we chose Known Pro-Establishment (also called Pro-Beijing) camp/Pro-Democracy camp and Unknown Pro-Establishment camp/Pro-Democracy camp. The contention between Pro-Establishment and Pro-Democracy is similar to the left/right division in the US and the two represent the

two extreme political stances in Hong Kong. The same directly parallel approach goes for all other statements in the project. For instance, we have the same false statements across the countries (merely changing, e.g., “Local government officials in Michigan” to “Local government officials from [a specific region in Poland]/[Hong Kong]” or changing “a new disease outbreak in Arizona, which threatens to take the lives of thousands of U.S. citizens” to an outbreak in a city in Poland or in Hong Kong). This careful selection and adaptation for both the treatment texts and other study materials assures that the politicians are equally known and partisan across the countries, that the sentences have the same baseline level of plausibility, and so forth. For that reason, the project entailed a collaboration of local social scientists who are based in the tested countries and intimately familiar with the local sociopolitical and cultural context.

In order to increase the transparency and allow for direct inspection and translation, we now include all these statements in English, Polish, and Cantonese in Appendix Subsections A2.1 - A2.3.

We hope that this fully addresses the reviewer’s query.

- With very little exception, in a subsequent wave of the US experiment, none of the treatments have any effects on misperceptions, skepticism, and trust (Table A12). One could argue that this means both the intended (misperception reduction) and unintended consequences (increases skepticism) are fairly short-lived.

We thank the reviewer for this comment. We now mention this in the main text (rather than in a footnote, as before) under the ‘General Effects’ section in the manuscript.

- The experiments are clear that almost all methods of misperception correction reduce the perceived accuracy of both true and false claims. However, one common method of examining the effects of intervention is using truth discernment - the perceived accuracy of true relative to false headlines (Pennycook & Rand, 2021). Even if the treatments have the unintended result of making people more skeptical even when they shouldn't be, do they still improve truth discernment?

Thank you for this important note. Following this comment, and the comment from Reviewer 4, we additionally created a subtractive measure that evaluated truth discernment. The subtractive truth discernment measure was created by subtracting the false statements measure (which tapped accuracy) from the true statements measure. Using this truth discernment measure as our outcome, we find that in the United States both accountability ($b = 0.45$, $p = 0.002$) and correctability ($b = 0.49$, $p = 0.001$) fact checking increased truth discernment while in Poland decontextualized media

coverage ($b = -.37, p = 0.01$) decreased truth discernment among participants. The treatments had no effect on truth discernment among Hong Kong participants. We have included the results of this measure in the Appendix. To stay close to our pre-analysis plan, we still focus our paper on the results of the evaluation of true/false claims separately. While we agree that the discernment measure offers valuable insights, such a measure does not account for the likely prevalence of true and false information in the real world: that is, a separate evaluation of how our interventions affect true and false claims tells us more about what is driving discernment. Given that there is more true than false news out there, even a small decrease in the perceived accuracy of true news is problematic. An improvement in discernment may therefore not necessarily be desirable. We hope that the reviewer understands our approach, and we thank the reviewer for suggesting to add this additional measure.

- The theoretical reasons for switching from a focus on falsity to a focus on bias make sense. However, given that doing so does not change the overall effect of exposure to media literacy material, it does raise the question over whether the public meaningfully distinguishes falsity from bias. This is worth discussing in the conclusion section.

We have included a discussion of this excellent point (underlined in the text below) in the conclusion section, which now reads: *“One direction is to make more efforts in targeted misinformation mitigation, so that the effective strategies we identified can successfully reduce misperceptions among vulnerable communities \cite{roozenbeek2020susceptibility}. The other direction is to rethink the current approaches and re-design messaging and delivery strategies (e.g., media literacy interventions) that do not inflate skepticism, for instance by focusing on that what is true rather than false \cite{acerbi2022research}. Our media literacy interventions, both the Bias and Misinformation focus, did not affect people's levels of misperceptions or skepticism. While there are theoretical reasons for shifting focus from falsity to bias in media literacy interventions, our findings raise the question of whether the public effectively distinguishes between concepts such as falsehoods and bias. Media literacy programs need to address these challenges not only by teaching the differences between falsity and bias but also by providing practical tools and strategies for identifying and evaluating both ---alongside balanced news --- in real-world contexts.”*

Similarly, one limitation of how the authors switched from a de-contextualized to a contextualized treatment highlighting the prevalence of false news is that there is no addition of a numerical anchor of fake news exposure. Since most people struggle with numeracy and intuitively processing small, but

non-zero percentages, it may be that even when people are told that fake news isn't a huge problem, that they are still greatly overestimating its prevalence and impact. While I don't think this is a fatal flaw in the paper, it might be a reason why contextualizing seems not to do much.

We agree with the reviewer that this indeed may be an explanation. We have included this observation as an avenue for future research to further strengthen interventions against misinformation. The section now reads:

“All in all, more research is needed to verify in which ways our proposed strategies may be further improved. For example, given that the differences between our treatments were subtle, future research could design strategies which differ more substantially from existing strategies. This way it may be harder to determine what aspect of the intervention is driving any effects, but ---arguably--- more substantial changes to existing interventions against misinformation are needed in order to maximize their benefits and limit their harms. For instance, researchers may more clearly contextualize the extent of misinformation in media coverage. Although many individuals struggle with numerical concepts, treatments that present people with a numerical anchor of fake news exposure may be more impactful. Without such an anchor, people may still overestimate the prevalence and impact of fake news, even when informed about their limited extent.”

PRESENTATION:

- The authors share several figures in the Appendix of the treatments. However, they are quite small and difficult to read. Since supplemental information can basically be of unlimited length online, there seems to be no need to keep them so small. Enlarging them would make it easier for attentive readers to follow.

Thank you for noting this. We now change the size of the figures so that they are easily visible and readable.

- The stimulus materials included appear to only be the ones in the US experiment. However, it would be worth adding the materials for the Polish and Hong Kong experiments.

Thank you for this note. We agree with the reviewer and add all the original study materials (the stimuli and other) from Hongkong and Poland to the Appendix. In the main manuscript, we included Figure 1 in order to provide a clear visual presentation and overview of all misinformation correction strategies, and Figure 4 in the method section to give the reader a clear picture of how the

treatments were presented (e.g., standardized number of comments and likes across conditions - and the background functions blurred). We used the US examples in the main manuscript so that it would be readable to the readership of NHB.

- While there are explanations for the preregistered outcomes in the OSF pre-registration for Tables A12-A24 in the Appendix, they are not present in the actual manuscript. Even if the authors do not wish to mention much of these results in analysis, some explanation within the manuscript would be helpful. That said, I do think Table A12 could be worth a mention as evidence that both intended and unintended consequences of misinformation correction are short-lived. Tables A13/A15 (US), A17/A19 (Poland), and A21/A23 (Hong Kong) provide useful evidence that the media literacy treatments, in practice, don't seem to help people with actually identifying false and biased headlines (respectively) in real time.

We thank the reviewer for this comment. We now mention the longevity of the effects in the US in-text (rather than in a footnote) under the 'General Effects' section in the manuscript. We included a footnote explaining that we focus on three pre-registered outcomes solely. The footnote reads: "For coherence and parsimony, we focus on the effects of each strategy on these three pre-registered outcomes. We report the results for the remaining pre-registered outcomes under Section \ref{others} in the Supplemental Information, and refer the reader to our pre-registration for the rationale of these remaining pre-registered outcomes."

REFERENCES:

Druckman, J.N., Levendusky, M.S., & McLain, A. (2018). No Need to Watch: How the Effects of Partisan Media Can Spread via Interpersonal Discussions. *American Journal of Political Science*, 62(1), 99-112. <https://dx.doi.org/10.1111/ajps.12325>

Pennycook, G., & Rand, D.G. (2021). The psychology of fake news. *Trends in Cognitive Sciences*, 25(5), P388-P402. <https://doi.org/10.1016/j.tics.2021.02.007>

Reviewer #4:

Remarks to the Author:

I was glad to have the opportunity to review this paper. The authors identify potential problems with existing approaches to tackling the problem of online misinformation, propose alternatives, and test both empirically.

I agree with the authors' assessment that judging misinformation interventions solely on their ability to decrease belief in (or sharing of) fake content is problematic, as they may also decrease belief in (or sharing of) real content. Given that there is far more real content than fake content in the information ecosystem, this appears extremely problematic. In general, I believe this is the strongest theoretical prediction and empirical result.

Thank you!

Much of the paper is concerned with testing the authors' concerns about specific misinformation intervention approaches and their proposed alternatives. I'm more skeptical of these critiques and proposed solutions.

1. Accountability vs. Correctability

- I understand how deemphasizing the source of the misinformation could lead to less distrust of news sources, but I don't think the authors are making a strong enough case in the current draft. There are currently only a few sentences on this in the paper,

We agree with the reviewer that our previous theoretical rationales may not have been sufficiently clear. We have expanded the relevant paragraph to emphasize that this strategy relies on rather subtle shifts in what is being emphasized in a correction. This paragraph now reads as follows:

"We propose that fact-checkers should consider what is important to emphasize when addressing (mis)information: the source (Accountability), or the verification of the relevant claim (which we call 'Correctability'). This strategy relies on fact-checkers' and journalists' expertise in issue and frame selection, in which they engage as part of their daily practice, and calls for an assessment of the need for Accountability versus the need for Correctability. More specifically, we suggest that --- when appropriate --- putting more emphasis on the false claim (as opposed to emphasizing the source of the claim) might overcome negative spillover effects on trust and misperceptions. By focusing on the correctability of a claim, fact-checkers can emphasize evidence-based thinking and critical analysis without attributing blame. This aligns with the broader goals of promoting media literacy and critical thinking skills, which are essential in a world inundated with information."

and I can't read the stimuli examples in the appendix due to the low resolution of the figures.

Thank you for directing our attention to this issue. We have improved the resolution of all figure in the Appendix as well as in the main manuscript. In addition, treatment texts can be found separately (written, not embedded in a figure) in the Appendix.

2. Misinformation vs. Bias Focus

- On “Bias focus”: “This should help citizens better evaluate the quality of information in general while at the same time reducing the negative effects on skepticism and trust by not overemphasizing the role of misinformation in the news media ecosystem.” But wouldn’t this Bias Focus similarly decrease trust and increase skepticism by emphasizing bias? This seems problematic.
- How does Bias Focus increase accuracy motivations? This doesn’t make sense to me. What is the mechanism there?
 - Relatedly, this line ---“Yet, only the ‘Bias Focus’ strategy should help identify accurate information”) ---confuses me...Are you saying that Misinformation Focus interventions do not help people to identify accuracy information. This is perplexing.

We thank the reviewer for raising these concerns, which were in part also mentioned by Reviewer 1. To this end, we have included an additional paragraph outlining four arguments why the bias focus should minimize increased skepticism in accurate information. This section now reads:

“Yet, only the ‘Bias Focus’ strategy should help identify accurate information without increasing skepticism towards all information for several reasons.

Because this strategy specifically teaches people to identify balanced or objective legacy media on top of biased media, it should help individuals identify accurate information. In addition, the Bias Focus should minimize skepticism towards all information because it may prompt individuals to think about how information is presented and framed. These tips should encourage people to evaluate the underlying assumptions and motivations behind news stories. This nuanced thinking should enhance media consumers' ability to identify not only reliable information or overt misinformation but also subtler forms of manipulation, such as selective reporting or framing. Furthermore, focusing on biases highlights the importance of context in news reporting. Media literacy interventions can teach individuals to look for multiple sources and perspectives to gain a more comprehensive understanding of an issue. This can naturally lead to the discovery of different viewpoints, without necessarily making individuals excessively skeptical of all information. Lastly, the Bias Focus still encourages critical thinking, but it shifts the emphasis from outright distrust to informed skepticism. Participants learn to evaluate news stories based on factors like source credibility, evidence, and logical coherence. Taken together, we propose that the Bias Focus empowers individuals to make more informed judgments

while consuming media.} In line with the arguments set out above, we hypothesize that participants exposed to the Bias \textit{or} Misinformation Focus will both be less likely to endorse misperceptions ($H2_{a}$), but only those exposed to the Bias Focus will have lower levels of skepticism ($H2_{b}$) and higher levels of trust ($H2_{c}$) than participants exposed to the Misinformation Focus.

3. De-Contextualized vs. Contextualized

- On a similar note, the contextualized alternative approach is confusing to me. My understanding of the research that the authors recommend journalists reference when covering misinformation shows that misinformation is not as large of a problem as commonly thought because it comes from a relatively small number of posters and is seen by relatively few people on social media. Yet the influence of misinformation can be far broader than this. It is covered by the media, spread through other channels, and can enter into political discourse (note that this paper begins by detailing the many ways in which misinformation is dangerous).
- Moreover, covering a piece of misinformation (e.g., microchips in vaccines) and then telling readers that concerns about misinformation are overblown seems like an odd way to cover misinformation, particularly given uncertainty about how overblown concerns about misinformation really are. Isn't this akin to telling people to get Covid booster shots while also mentioning that they aren't always deemed necessary for large segments of the population?
- One further point on the contextualized approach. You're suggesting that news outlets report on a piece of misinformation and then tell readers that concerns about misinformation are overblown, in part because not many people are exposed to misinformation. But by doing this, you're exposing people to misinformation. Given the illusory truth research that you reference, isn't this problematic?

We thank the reviewer for raising these critical, important points. We based our treatments (both the de-contextualized and the contextualized) on the current ways the media typically cover misinformation. More often than not, news media report on the problem of misinformation alongside reporting a falsehood. See for instance this news article: <https://www.vox.com/2018/4/18/17252410/jordan-peepe-obama-deepfake-buzzfeed>, where the original falsehood (in this case, a deepfake about Obama) is covered, but at the same time the article stresses the dangers of misinformation. We agree that it seems counterintuitive to suggest news media keep on repeating a falsehood when covering misinformation in our contextualized approach. However, news media cover misinformation in order to debunk false information about an event or what someone said and at the same time to raise awareness about the problem (in this

way, news coverage of misinformation differs from clear-cut fact-checks).

In fact, controversially, one approach and recommendation would be to have news media not cover misinformation at all. This approach cannot naturally be tested in our experiment, yet would ascertain that few people are exposed to misinformation in the first place. This approach, however, is not likely realistic. Although there certainly are news articles that do not address a particular falsehood at all and merely talk about the scope of misinformation, our aim was to think about a way media can keep correcting certain falsehoods, but without inducing skepticism.

There are, as the reviewer points out, of course ways in which misinformation can still be harmful. However, we suggest that when covering misinformation, news media put it in the context of the best available evidence we have about the true threats of misinformation. Such threats are of course not fixed, and there are many things we still don't know. We therefore think it is important to point out that 'the context' can change, and that news coverage of misinformation should adequately reflect this. While there are exceptions, we formulated the context based on the most recent knowledge we have about the effects of misinformation.

On a more technical note, in each single treatment, we aimed to only manipulate one aspect to see whether that would be driving/changing any effects. In the misinformation coverage treatments, this was only the context, not the absence or presence of a falsehood.

We hope the reviewer can agree with our considerations, and we thank the reviewer for letting us reflect on our treatments more clearly. Accordingly, we have revisited relevant paragraphs in the theory section and discussion. These changes are all highlighted in blue.

- Guay & colleagues have a short paper, forthcoming in Nature Human Behavior, on a similar topic. They advocate for measuring belief in both true (i.e., real) and false (i.e., fake) content, partially for the reasons you lay out in this paper. They also advocate for using a measurement of discernment as the primary outcome of interest. Essentially, the treatment effect on discernment captures how much more people in the treatment group believe true vs. false content than those in the control group. Right now you report the constituent parts of discernment (belief in false, disbelief in true), but really what you care about is whether the difference between them is significant, right? That might be worth considering.

Thank you for directing our attention to this paper, with which we were not familiar. The paper will certainly guide the design of our future studies. Meantime, in the analyses reported in our manuscript,

we stay close to the pre-registered analysis plan. Following this comment, and the comment from Reviewer 3, we additionally created a subtractive measure that evaluated truth discernment. The subtractive truth discernment measure was created by subtracting the false statements measure (which tapped accuracy) from the true statements measure. Using this truth discernment measure as our outcome, we find that in the United States both accountability ($b = 0.45, p = 0.002$) and correctability ($b = 0.49, p = 0.001$) fact checking increased truth discernment while in Poland decontextualized media coverage ($b = -.37, p = 0.01$) decreased truth discernment among participants. The treatments had no effect on truth discernment among Hong Kong participants. We have included the results of this measure in the Appendix. To stay close to our pre-analysis plan, we still focus our paper on the results of the evaluation of true/false claims separately. While we agree that the discernment measure offers valuable insights, such a measure does not account for the likely prevalence of true and false information in the real world: that is, a separate evaluation of how our interventions affect true and false claims tells us more about what is driving discernment. Given that there is more true than false news out there, even a small decrease in the perceived accuracy of true news may be more problematic than a larger decrease in the perceived accuracy of false news. An improvement in discernment may therefore not necessarily be desirable. We hope that the reviewer understands our approach, and we thank the reviewer for suggesting to add this additional measure.

- This paper contains a lot of labels, and they are sometimes hard to keep track of (the table helps with this). The accountability and correctability labels were hard to keep track of as a first time reader. The other labels are more self-explanatory.

Thank you for bringing our attention to this. We have now applied consistent labeling and capitalizing and hope this enables readers to keep track.

- I can't see Figure 2 clearly enough to tell difference between the interventions. I'm having the same issue for the stimuli figures in the appendix

Thank you for this important note, and we apologize for the inconvenience. We have enlarged and/or improved the resolution of all figures in the manuscript, including Appendix. That said, making Figure 2 (which is now Figure 1) larger such that one can properly read the text, would take up a lot of space, and hence is not really appropriate for the main text. We still believe that including this figure is helpful: it gives readers a clear visual of which but mostly how treatment materials were presented to participants (e.g., as a Facebook post on a timeline etc.). For that reason, we have kept the original Figure, but highlighted the purpose of this figure more clearly in the text. We hope this

approach meets the Reviewer's expectations.

We have now made sure that the figures (and all other materials presented in the final manuscript) are of high resolution and larger.

- Do you cluster standard errors, either at the respondent level (each respondent sees multiple headlines) and/or the headline level (each headline is seen by multiple respondents). The latter is only necessary if the same headlines were not seen by all respondents (that is, if all respondents saw the same X headlines, this isn't an issue). But either way you need to account for the nested structure of the data.

We thank the reviewer for this important suggestion. We now cluster standard errors at the participant level only (given that each participant saw the same true/false claims). The clustering of standard errors did not change any of the main results.

-There are some small spelling and grammatical errors throughout. Another careful read-through may be helpful.

Thank you for your close reading of the manuscript. We have now made sure that all the small spelling and grammatical errors are corrected.

- It would be nice to briefly mention why some strategies may be harmful in the introduction (even in a single sentence), rather than teasing them. The current version reads "We argue that the reason that these interventions may generate misperceptions, skepticism toward verified facts, and political and institutional distrust has to do with the way the message is delivered, as detailed below."

We agree with the reviewer and have added some examples of why interventions may be harmful.

The relevant sentences in the introduction now read: *"We argue that the reason that these interventions may generate misperceptions, skepticism toward verified facts, and political and institutional distrust has to do with the way the message is delivered, as detailed below. For instance, strategies against misinformation often adopt a negative tone (e.g., "Fight the Fake", "Proceed with Caution"), put blame on political actors and media outlets, or amplify the harms of misinformation."*

- I'm confused by this sentence: "After all, some evidence suggests that exposure to and effects of misinformation and untrustworthy sources are limited". Is your point just that exposure to misinfo doesn't really have much of an effect, so reporting on it in the news might have a worse effect? If so,

that's not coming through. Also, if so, I think you need to elaborate on the mechanism a bit. Is the idea that misinfo has a limited effect in the wild because people aren't very exposed to it, and that when news outlets cover it people are exposed to it more than they would otherwise have been?

Upon rereading the paragraph, we agree with the reviewer that it was somewhat underdeveloped. We have rewritten the paragraph and added several clarifications. The paragraph now reads:

*"Furthermore, news media coverage of misinformation often alarmingly emphasizes the existence, prevalence, spread, and threats of misinformation, disinformation, and untrustworthy sources \cite{altay2021misinformation}, without putting these threats in the necessary context. That is, some evidence suggests that exposure to and effects of misinformation and untrustworthy sources are limited \cite{aslett2022news, grinberg2019fake, guess2020fake, guess2019less, guess2020exposure}. \textcolor{blue}{When covering misinformation, news media may therefore not only overstate the risks but also inadvertently repeat false claims people would otherwise not have come across \cite{hoes2022cure}.} We call such coverage 'De-Contextualized approach' and suspect that it may have unintended consequences such that it decreases trust and increases skepticism toward verified facts \cite{hoes2022cure, van2023can}. Moreover, by typically repeating falsehoods, news coverage of misinformation may come with a risk of fostering misperceptions by making them more familiar \cite{zajonc1968attitudinal}, more easily retrievable from memory and therefore more easy to process \cite{swire2017processing}." **We hope this rewritten paragraph now meets the reviewers concerns.***

- You spend a lot of real estate in the paper explaining country-specific findings. Why? Can you motivate this a bit more?

Following Reviewer 1's suggestion, we have included a table summarizing country-specific results. Based on the suggestion above, we additionally shortened the two paragraphs on country-specific findings. We agree that it takes up some (but not too much) space, yet believe our study is valuable such that it tests interventions in various countries, some of which are not usually surveyed in misinformation research. We believe it would be a shame to not spend a bit of time going into detail on some of the differences between the US and Poland, without making it the main concern of the paper. We hope that with the table and the shortened paragraphs, the reviewer now feels the extent of the discussion of the country-specific findings are appropriate.

Very small comments:

- "democratic consequences" in abstract and intro reads funny. Consequences for democracy?
- "participants exposed to the bias and Misinformation Focus will be less likely to endorse Misperceptions". I had to read this part of the paper a few times, before realizing that there is no

condition that exposes people to both the Misinformation Focus and Bias Focus interventions. Also, it gets confusing when you use labels for some things (e.g., Misinformation Focus, in this sentence) but not others ("but those receiving a media literacy treatment focused on news bias"). Consistent labelling and capitalization helps the first time reader who is trying to remember which labels refer to what.

Thank you for directing our attention to these confusing expressions and and inconsistent labeling. We have adapted this throughout the manuscript.

- Figure 3 is the main plot, but is quite small. What if you vertically arrange the facets? Just an idea.

Great idea, thank you!

- Missing figure captions in most figures

Thank you for this comment. We have now corrected this oversight on our part and added figure captions to all the figures.

Decision Letter, first revision:

18th September 2023

Dear Emma,

RE: "Prominent Misinformation Interventions Reduce Misperceptions but Increase Skepticism"

Thank you for submitting your revised manuscript and for all your work on the revision.

Although your manuscript has been revised in response to reviewer comments, it does not fully comply with our editorial policies and formatting requirements. For example, statistical results are not fully reported or interpreted according to our requirements.

Before we can send the manuscript back to our reviewers, we ask that you revise it to ensure that it complies fully with our policies and is formatted according to our requirements. In particular, we ask that you address the following:

1) Fully report all inferential statistical results, including coefficients/effect sizes, p-values, and confidence intervals. In cases where you refer to tables in the Supplementary Information for results, these must contain full information, including both p-values and confidence intervals. Asterisks to

denote statistical significance should not be used as a substitute for exact p-values.

2) Ensure that you do not interpret marginally significant results (i.e. $p=0.05$ or above) as theoretically informative.

3) Ensure that you do not interpret null results as theoretically meaningful. For instance, the statement on pg. 10 "with the Correctability strategy ($b = -0.29$, $p = 0.001$) being slightly more successful at doing so than the Accountability strategy ($b = -0.21$, $p = 0.001$), although this difference is not statistically significant" should be revised to avoid implying that a difference between the two strategies was found.

4) Although not a requirement for further peer review, we recommend that you remove footnotes at this stage, in order to save time later. Our format does not permit footnotes, and these will ultimately need to be incorporated into the text or removed.

In order to assist with these revisions, I have attached another copy of our checklist. If you are uncertain as to how to address any of the above points, please don't hesitate to contact me.

[REDACTED]

Thank you in advance for attending to these requests and I look forward to receiving your revised manuscript.

Sincerely,
[REDACTED]

Decision Letter, second revision:

17th January 2024

Dear Dr. Hoes,

Thank you for your patience as we've prepared the guidelines for final submission of your Nature Human Behaviour manuscript, "Prominent Misinformation Interventions Reduce Misperceptions but Increase Skepticism" (NATHUMBEHAV-23051681B). Please carefully follow the step-by-step instructions provided in the attached file, and add a response in each row of the table to indicate the changes that you have made. Please also address the additional marked-up edits we have proposed within the reporting summary. Ensuring that each point is addressed will help to ensure that your revised manuscript can be swiftly handed over to our production team.

We would hope to receive your revised paper, with all of the requested files and forms within two-three weeks. Please get in contact with us if you anticipate delays.

Nature Human Behaviour offers a Transparent Peer Review option for new original research manuscripts submitted after December 1st, 2019. As part of this initiative, we encourage our authors to support increased transparency into the peer review process by agreeing to have the reviewer comments, author rebuttal letters, and editorial decision letters published as a Supplementary item. When you submit your final files please clearly state in your cover letter whether or not you would like to participate in this initiative. Please note that failure to state your preference will result in delays in accepting your manuscript for publication.

In recognition of the time and expertise our reviewers provide to Nature Human Behaviour's editorial process, we would like to formally acknowledge their contribution to the external peer review of your manuscript entitled "Prominent Misinformation Interventions Reduce Misperceptions but Increase Skepticism". For those reviewers who give their assent, we will be publishing their names alongside the published article.

Cover suggestions

We welcome submissions of artwork for consideration for our cover. For more information, please see our guide for cover artwork.

ORCID

Non-corresponding authors do not have to link their ORCIDs but are encouraged to do so. Please note that it will not be possible to add/modify ORCIDs at proof. Thus, please let your co-authors know that if they wish to have their ORCID added to the paper they must follow the procedure described in the following link prior to acceptance:

Nature Human Behaviour has now transitioned to a unified Rights Collection system which will allow our Author Services team to quickly and easily collect the rights and permissions required to publish your work. Approximately 10 days after your paper is formally accepted, you will receive an email in providing you with a link to complete the grant of rights. If your paper is eligible for Open Access, our Author Services team will also be in touch regarding any additional information that may be required to arrange payment for your article.

Please note that *Nature Human Behaviour* is a Transformative Journal (TJ). Authors may publish their research with us through the traditional subscription access route or make their paper immediately open access through payment of an article-processing charge (APC). Authors will not be required to make a final decision about access to their article until it has been accepted. Find out more about Transformative Journals

[REDACTED]

Best regards,
[REDACTED]

On behalf of

[REDACTED]

Reviewer #1:
Remarks to the Author:
The authors answered satisfactorily to my comments.

Reviewer #2:
Remarks to the Author:

Thank you for giving me the opportunity to review the revised version of this manuscript.
Overall, the authors have addressed all my comments, and the manuscript has improved significantly.

I have one suggestion: The country differences, especially the fact that in Hong Kong, skepticism did not increase with the 'improved strategies' is not addressed in the manuscript. I would be curious to read a potential explanation here.

Two minor points:

The name is Donsbach, not Donsbagh, 1996

And the labeling of the Tables is partly missing or incorrect, e.g. Table 1 is Table 2 (p. 8)

Good luck with your work!

Reviewer #3:

Remarks to the Author:

Many thanks to the authors for responding so thoughtfully to the suggestions of myself and the other reviewers. I am satisfied with the changes made and think this paper is in a great position to be cited in multiple disciplines.

Reviewer #4:

Remarks to the Author:

I am satisfied with the authors' responses and revisions to my original comments regarding their manuscript titled "Prominent Misinformation Interventions Reduce Misperceptions but Increase Skepticism." Given these revisions and my positive impression of the paper, I recommend publishing the paper in Nature Human Behavior.

Author Rebuttal, second revision

Reviewer 2

Thank you for giving me the opportunity to review the revised version of this manuscript.
Overall, the authors have addressed all my comments, and the manuscript has improved significantly.

I have one suggestion: The country differences, especially the fact that in Hong Kong, skepticism did not increase with the 'improved strategies' is not addressed in the manuscript. I would be curious to read a potential explanation here.

Two minor points:

The name is Donsbach, not Donsbagh, 1996

And the labeling of the Tables is partly missing or incorrect, e.g. Table 1 is Table 2 (p. 8)

Good luck with your work!

Response

We thank Reviewer 2 for their positive notes. Following the suggestion about the country differences, we have included the following paragraph below in the discussion section. We hope this meets the Reviewer's expectations. We also got rid of other typos and errors as rightly noted by the reviewer.

"Finally, we offer two potential explanations for the (lack of) findings for Hong Kong. First, the Hong Kong findings might be due to the sample age difference. We had difficulties in recruiting older adults from the Hong Kong survey vendor. In the Hong Kong sample, we only had 9.2% of participants aged 55 and older. This age category was 33.1% in the Poland sample and 31.9% in the US sample. The relatively younger sample in Hong Kong might have more baseline media literacy and experiences with misinformation corrections. This could potentially attenuate the intervention effects. Another possible explanation has to do with Hong Kong's political environment. Some scholars argue the lack of impact of misinformation or misinformation intervention is due to largely steady political inclinations of the Hong Kong population. The overall trend in public opinions regarding whether Hong Kong should be governed under the One Country, Two Systems has not changed much since 2014 or even earlier. Thus, regardless of frequent exposures to misinformation and increasing availability of misinformation interventions, the influence of these interventions might be limited \cite{kajimoto2022fact}"

Final Decision Letter:

Dear Emma,

I am happy to let you know that your Article "Prominent Misinformation Interventions Reduce Misperceptions but Increase Skepticism", has now been accepted for publication in *Nature Human Behaviour*.

Please note that *Nature Human Behaviour* is a Transformative Journal (TJ). Authors may publish their research with us through the traditional subscription access route or make their paper immediately open access through payment of an article-processing charge (APC). Authors will not be required to make a final decision about access to their article until it has been accepted. Find out more about Transformative Journals

With best regards,

[REDACTED]